# LEARNING GUARANTEE OF REWARD MODELING USING DEEP NEURAL NETWORKS

## ABSTRACT

In this work, we study the learning theory of reward modeling using pairwise comparison data and deep neural networks. We establish a novel non-asymptotic regret bound for deep reward estimators in a non-parametric setting, which depends explicitly on the network architecture. Furthermore, to underscore the critical importance of clear human beliefs, we introduce a margin-type condition requiring the conditional winning probability of the optimal action in pairwise comparisons to be significantly distanced from 1/2. This condition enables a sharper regret bound, which substantiates the empirical efficiency of Reinforcement Learning from Human Feedback and highlights the role of clear human beliefs in its success. Notably, this improvement stems from high-quality pairwise comparison data under the margin-type condition and is independent of the specific estimators used, making it applicable to various learning algorithms and models.

## 1 INTRODUCTION

Reinforcement Learning from Human Feedback (RLHF) has proven highly effective in aligning large language models with human preferences and expert policies (Christiano et al., 2017). A notable advancement in this field, Direct Preference Optimization (DPO), enhances RLHF by learning rewards directly from pairwise comparison data rather than environmental interactions. This approach significantly improves efficiency and achieves more robust alignment with human preferences (Rafailov et al., 2024), particularly in applications where preference feedback is naturally intuitive, such as recommendation systems and image generation.

The success of the RLHF has ignited extensive research to establish performance guarantees for learning optimal policies. For instance, Zhu et al. (2023) examines the theoretical properties of the reward modeling from action-based comparison data, while Chen et al. (2022) and Saha et al. (2023) focus on trajectory-based comparison data. However, the theoretical foundations of DPO remain largely unexplored, especially for deep neural network (DNN) estimators, creating a gap in understanding its full potential. Additionally, the reason why RLHF significantly improves sample efficiency compared to traditional RL methods remains less clear. Studies have considered the gap condition on the Q function to distinguish the optimal action from the others (Shi et al., 2023; Zhan et al., 2024). This provides a key insight into explaining the efficiency of RLHF based on reward modeling. Specifically, an underlying reward function structure is a plausible explanation for the observed efficiency gains in RLHF since the reward signals can always be reconstructed from the clear human preference feedback.

Strong consensus in pairwise comparison outcome reveals clear human preferences, indicating decisive winning odds for preferred actions and providing crucial insights into the underlying reward function for RLHF. These preferences are vital for aligning language models with complex psychological attributes, like honesty and harmlessness, that traditional reinforcement learning struggles to quantify. The clarity of these preferences enables RLHF to achieve remarkable performance with minimal pairwise data, even in highly complex environments, demonstrating its efficiency in reward modeling (Wang et al., 2024a). While recent work has explored structural conditions of the reward function to explain this efficiency (Shi et al., 2023; Rafailov et al., 2024), these conditions remain largely unverifiable, limiting their practical explanatory power.

In this paper, we focus on obtaining theoretical guarantees for reward models with well-designed network architectures. We aim to build a mathematical model to calibrate the clear preferences and explain the sample efficiency of RLHF based on reward modeling. Specifically, our contributions focus on the following aspects:

- We establish regret bounds for reward modeling using deep neural networks, which depend explicitly on network width, depth, and sample size. These bounds provide valuable insights into the success of DNN-based estimators in RLHF, particularly in terms of their sample efficiency.

- We propose a novel margin-type condition to calibrate clear human beliefs in RLHF. The condition implies high-quality pairwise comparison datasets and unveils the structure of the underlying reward, under which we obtain a sharper regret bound. This finding highlights the role of clear human beliefs in its success. The theoretical improvements are independent of the estimators used in practice, allowing them to be applied across a wide range of learning algorithms.

- We emphasize the broad applicability of the theoretical results in our work. Our findings provide theoretical guarantees for removing ambiguous comparison data during the preprocessing stage of RLHF training. We examine DNN-based reward estimators under general pairwise comparison models, without restricting them to specific parameterizations, thereby supporting RLHF's empirical efficiency across various scenarios.

The rest of the paper is organized as follows. In Section 2, we introduce the pairwise comparison model and propose the margin-type condition. We demonstrate that this condition leads to faster convergence of the resulting regret. In Section 3, we derive non-asymptotic regret bounds for deep reward estimators, which are explicitly characterized by the structure of DNNs. We also discuss the implications of these results for generalization. We review related literature in Section 4 and conclude with future research directions in Section 5. Technical details are deferred to the Appendix.

## 2 PAIRWISE COMPARISON, MARGIN-TYPE CONDITION AND SHARPER BOUND

In this work, we consider the reward modeling in the action-based pairwise comparison case. Let $\mathcal{S}$ be the set of states (prompts), and $\mathcal{A}$ be the set of actions (responses). We consider a pairwise comparison dataset $\{s^i, a_1^i, a_0^i, y^i\}_{i=1}^N$ with sample size $N$: the state $s^i$ is sampled from the probability measure on state space $\mathcal{S}$, denoted by $\rho_s$; Conditioning on the state $s^i$, the action pair $(a_1^i, a_0^i)$ are sampled form some joint distribution $\mathbb{P}(a_1, a_0|s^i)$; The comparison outcome $y^i$ indicates the preference between $a_1^i$ and $a_0^i$. Specifically, $a_1^i$ is preferred over $a_0^i$ if $y^i > 0$ and conversely, $a_0^i$ is preferred if $y^i < 0$. It is worth noting that we do not restrict the outcome to a binary format, and it accommodates various types of outcomes discussed in the literature. For simplicity, readers may consider the binary case where $y^i$ takes on values of either $-1$ or $1$. We define the reward function $r : \mathcal{S} \times \mathcal{A} \to \mathbb{R}$, which evaluates the reward of taking each action at a given state. We denote $d$ as the dimension of the input for reward function $r$.

For any reward function $r$, we denote the decision maker by $\pi_r(s) = \arg\max_{a \in \mathcal{A}} r(s, a)$. Let $r^*(s, a)$ denote the underlying optimal reward function, we define the optimal action for the state $s$ by $\pi_{r^*}(s) = \arg\max_{a \in \mathcal{A}} r^*(s, a)$. For an estimated reward function $r$, we are interested in the regret of the induced $\pi_r(s)$, which is

$$\mathcal{E}(r) = \int_{\mathcal{S}} r^*(s, \pi_{r^*}(s)) - r^*(s, \pi_r(s))d\rho_s. \tag{1}$$

The regret (1) is an intrinsic measure for evaluating RLHF (Zhu et al., 2023; Zhan et al., 2024). It is important to note that the policy $\pi_r$ is induced from the reward $r$ and the resulting regret is determined by $r$. In practice, we optimize the comparison model using crowd-sourced comparison outcomes. Even if the reward function is not perfectly estimated, there remains an opportunity to derive a correct policy and achieve low regret in reinforcement learning tasks. This potential stems from clear human beliefs, which suggest a significant gap between the reward of the optimal action and its alternatives. However, the critical role of reward differences between actions is often overlooked in pairwise comparison analysis, which typically relies on the smoothness of the reward

function. In light of this, we are motivated to quantify the effects of reward differences on regret, capturing the reward gap between actions.

As discussed in Wang et al. (2024a); Song et al. (2024); Zhan et al. (2024), we model the relationship between comparison response and the difference of the rewards $r^*(s, a_1) - r^*(s, a_0)$. Specifically, the probability of the event that $a_1$ is preferred over $a_0$ under the state $s$ can be expressed as:

$$\mathbb{P}(y > 0 \mid s, a_1, a_0) = \int_0^\infty g(y, r^*(s, a_1) - r^*(s, a_0))dy, \tag{2}$$

where the function $g$ represents the probability density function of the comparison outcome $y$ and in this paper we consider a general parametrization for $g$. It is worth noting that the success of RLHF is largely attributed to clear human preferences, while incorrect or ambiguous preference labels can lead to significant performance deterioration in practice (Saha et al., 2023; Wang et al., 2024a; Chen et al., 2024). To calibrate the clear human preferences, we propose the following margin condition in the pairwise comparison dataset.

**Assumption 1 (Margin Condition for the Human Preference)** *For any action pairs $(\pi_{r^*}(s), a')$ where $a' \in \mathcal{A} \setminus \pi_{r^*}(s)$ and $t \in (0, 1/2)$, we have*

$$\mathbb{P}_{\mathcal{S}}\left(\mathbb{P}(y > 0 \mid s, a_1 = \pi_{r^*}(s), a_0 = a') - \frac{1}{2} \le t\right)$$

$$:= \int_{\mathcal{S}} \mathbf{1}\left\{\int_0^\infty g(y, r^*(s, \pi_{r^*}(s)) - r^*(s, a'))dy - \frac{1}{2} \le t\right\}d\rho_s \le c_g t^{\frac{\alpha}{1-\alpha}},$$

*where $c_g > 0$ is a universal constant and $\alpha \in (0, 1)$ is the coefficients for quantifying the clear human belief. The larger $\alpha$ indicates a clearer preference in the pairwise comparison dataset.*

Assumption 1 implies that experts have a clear tendency between the optimal action and the other for most states $s$, under which the winning probability of the optimal action is bounding away from 1/2. It is worth noting that $\alpha = 0$ and $\alpha = 1$ correspond to two extreme cases respectively for the case without any margin-type assumption and the noiseless case. To better understand Assumption 1, we take a closer look at its implication on the underlying reward function. Here, we present two classical comparison models with $y \in \{-1, 1\}$ as examples.

**Example 1 (BT model (Bradley & Terry, 1952))** *The comparison function is*

$$g(y, u) = \mathbf{1}(y = 1) \cdot \frac{\exp(u)}{1 + \exp(u)} + \mathbf{1}(y = -1) \cdot \frac{\exp(-u)}{1 + \exp(-u)}.$$

*Given a particular state-action pair, the probability of observing the outcome $y > 0$ is $\exp(r^*(s, a_1) - r^*(s, a_0))/(1 + \exp(r^*(s, a_1) - r^*(s, a_0)))$.*

**Example 2 (Thurstonian model (Thurstone, 1927))** *The comparison function is*

$$g(y, u) = \mathbf{1}(y = 1) \cdot \Phi(u) + \mathbf{1}(y = -1) \cdot (1 - \Phi(u)),$$

*where $\Phi(u)$ is the cumulative distribution function of the standard normal distribution. Then we have $\mathbb{P}(y > 0 \mid s, a_1, a_0) = \Phi(r^*(s, a_1) - r^*(s, a_0))$.*

The *BT model* and *Thurstonian model* are widely considered in RLHF and DPO modeling (Christiano et al., 2017; Rafailov et al., 2024; Siththaranjan et al., 2024). Besides that, there are some other comparison models used in DPO. For example, the Rao-Kupper model and Davidson model are employed to tackle pairwise comparisons with ties (abstentions), where $y$ takes values from $\{-1, 0, 1\}$ (Chen et al., 2024; Rao & Kupper, 1967; Davidson, 1970). Notably, all of these models are incorporated into the general comparison framework discussed in our work.

The comparison model connects the underlying reward function to human preferences within the observed comparison datasets. By considering the clear preference data outlined in Assumption 1, we could further reveal the specific structure of the reward function, which is summarized in the following Lemma 1.

**Lemma 1** *Given Assumption 1, with $\alpha \in (0, 1)$ and $t \in (0, c_{r^*})$ where $c_{r^*}$ is the upper bound of the true reward defined in Assumption 2, there exists a universal constant $c'_g$ such that*

$$\int_{\mathcal{S}} \mathbf{1} \left\{ r^*(s, \pi_{r^*}(s)) - \max_{a \in \mathcal{A} \setminus \pi_{r^*}(s)} r^*(s, a) \le t \right\} d\rho_s \le c'_g t^{\frac{\alpha}{1-\alpha}} .$$

*Specifically, $c'_g = (1/4)^{\alpha/(1-\alpha)} c_g$ in the BT model and $c'_g = (1/2\pi)^{\alpha/(2-2\alpha)} c_g$ in the Thurstonian model.*

Lemma 1 implies that a clear preference in the comparison dataset is determined by the reward margin between the two actions. Its validity also depends on the properties of the comparison function specified in Definition 1, making it applicable to the general comparison model. Unlike existing literature, which imposes unverifiable conditions directly on the reward structure, Lemma 1 is derived from Assumption 1 on high-quality preference datasets. This approach is more valid and enjoys greater generalization ability compared to existing conditions (Kim et al., 2021; Shi et al., 2023; Zhan et al., 2024).

## 2.1 SHARPER REGRET BOUND

In this section, we present the regret bounds of the decision maker $\pi_r(s)$ defined in (1) with and without Assumption 1.

**Theorem 1 (Faster Rate with Margin Condition)** *Let $r$ be some reward function estimator, with Assumption 1 holds and the margin parameter $\alpha \in (0, 1)$, there exist a universal constant $c_1 > 0$, such that*

$$\mathcal{E}(r) \le c_1 \left( \|r - r^*\|^2_{L^2(\mathcal{S}, \ell^2)} \right)^{\frac{1}{3-2\alpha}} ,$$

*where the norm $\| \cdot \|_{L^2(\mathcal{S}, \ell^2)}$ is defined in (6).*

Theorem 1 suggests that when a hard margin is imposed, i.e., $\alpha \to 1$, the regret of the "greedy" policy induced by the estimated reward function is at the order of $\mathcal{O}(\|r - r^*\|^2_{L^2(\mathcal{S}, \ell^2)})$, which is theoretically tight in the sense that the rate cannot be improved without additional conditions.

**Corollary 1 (Regret Bound without Margin Condition)** *Let $r$ be some reward function estimator. There exists a universal constant $c_2 > 0$, such that*

$$\mathcal{E}(r) \le c_2 \left( \|r - r^*\|^2_{L^2(\mathcal{S}, \ell^2)} \right)^{\frac{1}{3}} .$$

Theorem 1 first demonstrates the role of clear preference in the comparison dataset and on the regret bound (Zhu et al., 2023; Wang et al., 2024a). Comparing Theorem 1 and Corollary 1, we show that the regret bound can be improved significantly with the margin-type condition. It is worth noting that, the margin parameter $\alpha$ interpolates the regret bound under two extreme cases as our result reduces to the rate $\mathcal{O}((\|r - r^*\|^2_{L^2(\mathcal{S}, \ell^2)})^{1/3})$ if $\alpha = 0$ (no margin-type condition is imposed). The efficiency gain in our results adjusts automatically with the margin parameter $\alpha$ while remaining independent of the error $\|r - r^*\|^2_{L^2(\mathcal{S}, \ell^2)}$. This improvement is primarily attributed to the use of a high-quality pairwise comparison dataset and is universally applicable to any estimator $r$ employed (Audibert & Tsybakov, 2007; Kim et al., 2021). These findings are coherent with the empirical observations in the RLHF training (Wang et al., 2024a; Chen et al., 2024). We also obtain the convergence rate of action selection consistency in Section C.2 of the Appendix, which is a beneficial complement for us to further understand the effects of the margin-type condition.

Based on the margin-type condition, we now present an overview of our main result regarding the regret bound using DNN-based reward estimators.

**Theorem 2 (Informal, Guarantee for Deep Reward Modeling)** *Consider the deep reward estimator $\hat{r} \in \mathcal{F}_{\mathrm{DNN}}$ where $\mathcal{F}_{\mathrm{DNN}}$ is a class of the deep neural networks with width $W = \mathcal{O}(d^\beta)$ and depth $D = \mathcal{O}(\sqrt{N})$. Then under some regular Assumptions, with probability at least $1 - \delta$,*

$$\mathcal{E}(\hat{r}) = \mathcal{O} \left( \left\{ d^\beta N^{-\frac{\beta}{d+2\beta}} + \sqrt{\frac{\log(1/\delta)}{N}} \right\}^{\frac{1}{3-2\alpha}} \right) ,$$

where $\beta$ is the Hölder smoothness parameter for the reward function $r^*$ and $d$ is the dimension of the input $\mathcal{S} \times \mathcal{A}$.

Theorem 2 establishes the convergence rate for the regret of deep reward estimators in a fully non-parametric setting. By setting proper network depth $D$ and width $W$, the regret of deep reward estimator achieves a rate of $\mathcal{O}(N^{-\beta/[(d+2\beta)(3-2\alpha)]})$. Our analysis provides implications for practitioners on how to choose the neural network parameters and construct high-quality comparison datasets to achieve effective reward modeling.

## 3 LEARNING GUARANTEE OF DEEP REWARD MODELING

The log-likelihood function for $r$ on the pairwise comparison dataset is written as follows,

$$l(r) = \mathbb{E}\left[\log g\left(y, r(s, a_1) - r(s, a_0)\right)\right].$$

Correspondingly, the empirical log-likelihood is written as,

$$\hat{l}(r) = \frac{1}{N}\sum_{i=1}^{N}\log g\left(y^i; r(s^i, a_1^i) - r(s^i, a_0^i)\right).$$

For a given reward function $r$, the empirical risk $\hat{l}(r)$ is calculated using the observed pairwise comparison data, while the population risk $l(r)$ is the expected value of the risk. Given the pairwise comparison dataset, we obtain $\hat{r}$ with the following objectives,

$$\hat{r} \in \arg\max_{r \in \mathcal{F}_{\text{DNN}}} \hat{l}(r). \tag{3}$$

To establish the theoretical guarantee of the above estimator, several factors should be considered. First, the characteristic of the comparison function $g(y, u)$ captures the relationships between human preference and the underlying reward. Second, the smoothness of the true reward function $r^*$ determines how well it can be approximated. Most importantly, the influence of the neural network configurations, *i.e.*, depth and width, need to be leveraged, as it dictates the model's capacity and efficiency to learn complex patterns from finite samples. In the following, we provide definitions and assumptions related to these factors and shape the efficacy of data-driven reward modeling.

**Definition 1 (Comparison Function)** *A function $g : \Omega \times \mathbb{R} \to \mathbb{R}^+$, where $\Omega$ is a symmetric subset of $\mathbb{R}$ denoting the possible comparison outcomes, is said to be a comparison function if:*

*(i) For $u \in \mathbb{R}$, $\int_{\Omega} g(y, u)dy = 1$ if $\Omega$ is continuous, and $\sum_{y \in \Omega} g(y, u) = 1$ if $\Omega$ is discrete;*

*(ii) $g(y, u) = g(-y, -u)$, for any $(y, u) \in \Omega \times \mathbb{R}$;*

*(iii) For $y < 0, g(y, u)$ is decreasing with respect to $u$, and $g(y, u) \to 0$ as $u \to \infty$;*

*(iv) $\sup_{u \in \mathbb{R}} g(y, u) < +\infty$, for every $y \in \Omega$.*

*(v) For every $y$, $\partial^2 \log g(y, u)/\partial u^2 < 0$.*

These conditions ensure $g$ is a proper probability function with a symmetric preference structure, stronger preferences for larger relative scores, and log-concavity in $u$. These conditions are mild and widely considered in the literature. It is straightforward to check many commonly used models satisfy these conditions, including *BT model*, *Thurstonian model*, *Rao-Kupper model*, *Davidson model* and the *paired cardinal model* proposed in Shah et al. (2016). To proceed, we describe the characteristics of the reward functions in preference learning.

**Definition 2 (Hölder Function Class)** *For $\beta, c_{\mathcal{H}} > 0$, and a domain $\mathcal{X} \in \mathbb{R}^d$, the Hölder function class $\mathcal{H}^{\beta}(\mathcal{X}, c_{\mathcal{H}})$ is defined by*

$$\mathcal{H}^{\beta}(\mathcal{X}, c_{\mathcal{H}}) = \left\{f : \mathcal{X} \to \mathbb{R}, \max_{\|\boldsymbol{\omega}\|_1 \leq \lfloor\beta\rfloor}\|\partial^{\boldsymbol{\omega}}f\|_{\infty} \leq c_{\mathcal{H}}, \max_{\|\boldsymbol{\omega}\|_1 = \lfloor\beta\rfloor}\sup_{x \neq x'}\frac{|\partial^{\boldsymbol{\omega}}f(x) - \partial^{\boldsymbol{\omega}}f(x')|}{\|x - x'\|_2^{\beta - \lfloor\beta\rfloor}} \leq c_{\mathcal{H}}\right\},$$

*where $\boldsymbol{\omega} = (\omega_1, \ldots, \omega_d)^{\top}$ is a vector of non-negative integers, $\|\boldsymbol{\omega}\|_1 := \sum_{i=1}^{d}\omega_i$, and $\partial^{\boldsymbol{\omega}} = \partial^{\omega_1}\cdots\partial^{\omega_d}$ denotes the partial derivative operator.*

**Assumption 2** *The range of the target reward function $c_{r^*} := \max_{a \in \mathcal{A}} \sup_{s \in \mathcal{S}} r^*(s, a)$ is finite.*

**Assumption 3** *(i) The marginal probability measure $\rho_s$ is absolutely continuous with respect to the Lebesgue measure; (ii) For every $a \in \mathcal{A}$, the reward function $r^*(s, a)$ belongs to the Hölder class $\mathcal{H}^\beta([0, 1]^d, c_{\mathcal{H}})$ for a given smoothness parameter $\beta > 0$ and a finite constant $c_{\mathcal{H}} > 0$.*

To ensure the identifiability of $r^*$, we assume the reward function $\sum_{a \in \mathcal{A}} r^*(s, a) = 0$ for all $s$. It is worth noting that the assumption is more of a normalization condition instead of a constraint, as the winning probability is invariant to the shift of reward functions. Similar conditions are considered in Zhu et al. (2023); Rafailov et al. (2024). In addition, we mention that our theory is general and applies to any underlying reward function, not just normalized ones. For an unknown reward function, we can always transform it into a normalized version, and both the unnormalized and normalized functions lead to the same preference distribution (Rafailov et al., 2024). Also, we estimate the true reward with the condition $\sum_{a \in \mathcal{A}} \hat{r}(s, a) = 0$ for all $s$. In our theoretical study, the reward estimator is implemented by a fully connected feed-forward neural network consisting of multiple layers of interconnected neurons. Its structure can be described as a composition of linear mappings and activation functions. Specifically, we consider the class of functions $\mathcal{F}_{\text{DNN}}$ consists of $D$-layer feed-forward neural networks that can be expressed as follows,

$$r(s, a; \theta) = f_{D+1} \circ f_D \circ \cdots \circ f_2 \circ f_1(s, a), \tag{4}$$

where $f_i(x) = \sigma^{(i)}(H^{(i)} x + b^{(i)})$ is the transformation for layer $i$. $H^{(i)}$ and $b^{(i)}$ are the weight matrix and bias vector, respectively. $\sigma^{(i)}$ denotes the ReLU activation function, which is applied to its input elementwise. We denote the width of the neural network as $W$, which is the maximum of the width of all layers. Let $\theta = (H^{(1)}, b^{(1)}, \ldots, H^{(D+1)}, b^{(D+1)})$ represents all the parameters in the neural networks, which consists of $p$ entries in total.

### 3.1 ESTIMATION WITHIN DEEP NEURAL NETWORK FUNCTION CLASS

With the aforementioned specifications, we start our analysis with the excess risk, which in general stems from two sources: the error from random data realizations and the error from the DNN's limited capacity to represent the target reward. We formalize these intuitions in the following lemma.

**Lemma 2 (Excess risk decomposition)** *The excess risk of $\hat{r}$ is defined and decomposed as*

$$l(r^*) - l(\hat{r}) \leq 2 \sup_{r \in \mathcal{F}_{\text{DNN}}} |l(r) - \hat{l}(r)| + \inf_{r \in \mathcal{F}_{\text{DNN}}} [l(r^*) - l(r)]. \tag{5}$$

The first term of the right-hand side is the stochastic error, which measures the difference between the risk $l$ and the empirical counterpart $\hat{l}$ defined over function class $\mathcal{F}_{\text{DNN}}$, evaluating the estimation uncertainty caused by the finite sample size. The second term is the approximation error, which measures how well the function $r^*$ can be approximated using $\mathcal{F}_{\text{DNN}}$ with respect to the likelihood $l(\cdot)$. To assist the following analysis, we define the constants depending on $g(y, u)$ and the range $c_{r^*}$:

$$\kappa_0 := \sup_{y \in \Omega, |u| \leq c_{r^*}} |\log g(y, u)|,$$

and

$$\kappa_1 = \sup_{y \in \Omega, |u| \leq c_{r^*}} \left| \frac{\partial}{\partial u} \log g(y, u) \right|, \quad \kappa_2 := \inf_{y \in \Omega, |u| \leq c_{r^*}} \left| \frac{\partial^2}{\partial u^2} \log g(y, u) \right|.$$

These constants are used in the results, specifying the Lipschitz property and log-concavity invoked from Definition 1 that ensure the convergence of the maximum likelihood estimator from the deep neural network function class.

**Proposition 1 (Stochastic Error Bound)** *Under Assumption 2, there exists a universal constant $c_3 > 0$, with probability at least $1 - \delta$,*

$$\sup_{r \in \mathcal{F}_{\text{DNN}}} \left| l(r) - \hat{l}(r) \right| \leq \kappa_0 \sqrt{\frac{2}{N}} \left( c_3 \sqrt{|\mathcal{A}| D p \log \left( \frac{W((D+1)|\mathcal{A}|N)^{1/D}}{(W!)^{1/p}} \right)} + \sqrt{\log(1/\delta)} \right).$$

If $r^* \in \mathcal{F}_{\text{DNN}}$, Proposition 1 describes the additional cost of the in-sample learned reward function in terms of the likelihood functional compared to the optimal oracle, which scales as $\mathcal{O}(\sqrt{\log(N)/N})$ with well-designed network structures form the class $\mathcal{F}_{\text{DNN}}$. It is reasonable that with more collected samples, the DNN can learn the underlying reward function better. Meanwhile, the stochastic error bound increases with the complexity of the function class $\mathcal{F}_{\text{DNN}}$. In other words, once we already know that $r^* \in \mathcal{F}_{\text{DNN}}$ for some network parameters, there is no need to further increase the network's width and depth given the available samples.

**Proposition 2 (Approximation Error Bound)** *Let $\mathcal{F}_{\text{DNN}}$ be the deep ReLU neural network class with width and depth, respectively, specified as $W = 38(\lfloor\beta\rfloor + 1)^2 d^{\lfloor\beta\rfloor+1} M_1 \lceil\log_2(8M_1)\rceil$ and $D = 21(\lfloor\beta\rfloor + 1)^2 M_2 \lceil\log_2(8M_2)\rceil$. Under Assumptions 2 and 3, for any $M_1, M_2 \in \mathbb{N}^+$, we have*

$$\inf_{r \in \mathcal{F}_{\text{DNN}}} l(r^*) - l(r) \leq 36\kappa_1 c_{\mathcal{H}}(\lfloor\beta\rfloor + 1)^2 d^{\lfloor\beta\rfloor+\frac{(\beta \vee 1)}{2}} (M_1 M_2)^{-\frac{2\beta}{d}}.$$

Proposition 2 demonstrates that the approximation error bound is decreasing in the size of the function class $\mathcal{F}_{\text{DNN}}$ through two parameters $M_1$ and $M_2$, which are assigned later. This is intuitive since a larger network has greater expressive power. On the other hand, a larger network inflates the stochastic error due to the over-parameterization.

Consequently, it is necessary to design the network structure carefully to strike a balance between the stochastic error and the approximation error. To do this, we need to appropriately relate $M_1$ and $M_2$ to the sample size $N$ so that an optimal convergence rate of excess risk bound can be achieved. There are many approaches that result in the same optimal convergence rate while incurring different total numbers of parameters $p$. Note that $p \leq W(d + 1) + (W^2 + W)(D - 1) + W + 1 = \mathcal{O}(W^2 D)$, which grows linearly in the depth $D$ and quadratically in the width $W$. It is desirable to employ fewer model parameters, and thus, the deep network architectures are preferable to the wide ones.

To ensure the validity of the MLE estimator, we consider the graph structure of the pairwise comparison dataset, which requires the number of comparisons between all the action pairs not to be scarce.

**Assumption 4 (Data Coverage Assumption)** *Let $\Lambda \in \mathbb{R}^{|\mathcal{A}|\times|\mathcal{A}|}$ be the Laplacian matrix , where $\Lambda_{ij} = -n_{ij}/N$ and $\Lambda_{ii} = \sum_j n_{ii}/N$. $n_{ij}$ is the fraction of sample size that the pair $(a_i, a_j)$ is compared and $n_{ii}$ is the fraction of comparisons that $a_i$ involves. We assume that there exists a positive constant $\kappa_\Lambda$, such that $\lambda_2(\Lambda) > \kappa_\Lambda$ where $\lambda_2(\Lambda) = argmin_{w \perp \mathbf{1}} w^\top \Lambda w / w^\top w$.*

If some actions are almost not queried among the data, the spectral gap $1/\lambda_2(\Lambda)$ diverges, and thus the regret of the MLE estimator could not converge (Zhu et al., 2023, Theorem 3.9). We refer to Appendix A for more discussion and an Example, where $1/\lambda_2(\Lambda)$ can blow up, and the regret bound is not guaranteed consequently.

Next, we obtain the optimal bound in terms of the norm $\|\cdot\|^2_{L^2(\mathcal{S},\ell^2)}$, which is defined in (6).

**Theorem 3 (Non-asymptotic Estimation Error Bound)** *Given the network parameters being specified as in Proposition 2, with $M_1 = 1$ and $M_2 = \lfloor N^{d/(2d+4\beta)} \rfloor$. Under Assumption 2 - 4, there exists a universal constant $c_4 > 0$, with probability at least $1 - \delta$,*

$$\|\hat{r} - r^*\|^2_{L^2(\mathcal{S},\ell^2)} \leq 2\sqrt{2}\frac{\kappa_0}{\kappa_2\kappa_\Lambda}\left(c_4\sqrt{|\mathcal{A}|}(\lfloor\beta\rfloor + 1)^4 d^{\lfloor\beta\rfloor+1}(\log(N))^2 N^{-\frac{\beta}{d+2\beta}} + \sqrt{\frac{\log(1/\delta)}{N}}\right).$$

Theorem 3 presents the non-asymptotic convergence rate of $\|\hat{r} - r^*\|^2_{L^2(\mathcal{S},\ell^2)}$. Unlike deep regression and classification problems, the convergence of excess risk does not directly guarantee the functional convergence of the estimated reward unless additional constraints on the dataset structure are imposed (see Assumption 4). This condition is also considered by Zhu et al. (2023). To the best of our knowledge, we are the first to investigate its role in the non-parametric analysis of deep reward modeling. Combining Theorems 1 and 3, we obtain the regret bound for the decision maker $\pi_{\hat{r}}(s)$ induced by the deep reward estimator under the margin-type condition.

**Theorem 4** *Let $\mathcal{F}_{\text{DNN}}$ be the deep ReLU neural networks class with width and depth, respectively, specified as*

$$W = 114(\lfloor\beta\rfloor + 1)^2 d^{\lfloor\beta\rfloor+1} \quad \text{and} \quad D = 21(\lfloor\beta\rfloor + 1)^2 N^{\frac{d}{2d+4\beta}} \lceil\log_2(8N^{\frac{d}{2d+4\beta}})\rceil.$$

*Given the MLE estimator $\hat{r} \in \mathcal{F}_{\text{DNN}}$, under Assumptions 1 - 3, there exists a universal constant $c_5$, such that with probability at least $1 - \delta$,*

$$\mathcal{E}(\hat{r}) \leq c_5 \left( \frac{\kappa_0 \sqrt{|\mathcal{A}|}(\lfloor\beta\rfloor + 1)^4 d^{\lfloor\beta\rfloor+1}(\log N)^2}{\kappa_2 \kappa_\Lambda} \right)^{\frac{1}{3-2\alpha}} N^{-\frac{\beta}{(d+2\beta)(3-2\alpha)}} + \left( \frac{\kappa_0^2 \log\left(\frac{1}{\delta}\right)}{\kappa_2^2 \kappa_\Lambda^2 N} \right)^{\frac{1}{2(3-2\alpha)}}.$$

Our results reveal that deep neural network reward estimators offer satisfactory solutions with explicit theoretical guarantees in this general setting. To achieve the fast convergence rate, it is essential to train the model with sufficient data and to select an appropriate network structure, adhering to the guidelines for width and depth selection provided in Theorem 4. To be specific, the width is a multiple of $d^{\lfloor\beta\rfloor+1}$, a polynomial of the feature dimension $d$; The depth is proportional to $\sqrt{N}$, as $d \gg \beta$. This explicitness offers more informative insights than those characterized solely by network size. In Section E of the Appendix, we give an experiment on synthetic data to illustrate the applicability of our theoretical results for reward modeling with deep neural networks. Theorem 3 relies on the reduced complexity of neural networks, which is obtained through the lens of functional equivalence (Shen, 2024). Such functional equivalence generally holds for fully connected networks, Residual networks, and attention-based networks. Recently, Takakura & Suzuki (2023) derived a polynomial convergence rate for sequence-to-sequence learning tasks using transformers, and specifically, they examine the shift-equivariant properties of transformers. It would be of interest for future studies to extend the analysis in this paper to state-of-the-art architectures such as Bidirectional Encoder Representations from Transformers (BERT), Generative Pre-trained Transformers (GPT), and other attention-based models.

## 4 RELATED WORKS

**Reinforcement Learning from Human Feedback** Human preferences have emerged as a valuable alternative to numerical rewards in reinforcement learning (RL), owing to their intuitive elicitation process (Ziegler et al., 2019; Ouyang et al., 2022). Recent theoretical advances in offline RLHF have been made by Zhu et al. (2023) and Zhan et al. (2024), who developed reward-based preference models—the former focusing on linear models and the latter extending to a general function class. While these foundational works establish theoretical frameworks, the critical role of human belief quality in RLHF has only recently gained attention. Wang et al. (2024a) addresses this by developing protocols for preference label correction and ambiguity smoothing, while Zhong et al. (2024) tackles preference heterogeneity through meta-learning approaches. However, a rigorous theoretical framework for quantifying human belief in RLHF remains an open challenge.

**Margin Condition for Human Preference** Modeling The efficiency of RLHF is largely attributed to strong human preferences reflected in pairwise comparison data (Zhong et al., 2024; Wang et al., 2024a). This concept parallels established noise conditions in classification theory, particularly Massart and Tsybakov noise conditions, which bound excess misclassification error (Tsybakov, 2004; Diakonikolas et al., 2021; 2022). Similar margin-type conditions have proven valuable across various domains, from individualized treatment analysis (Qian & Murphy, 2011) to optimal policy identification in offline RL (Shi et al., 2023), and linear bandit problems (Goldenshluger & Zeevi, 2013; Bastani & Bayati, 2020). While recent empirical studies aim to enhance data quality and address ambiguous samples in practical applications, offering various treatment strategies for handling crowd-sourced preference data (Wang et al., 2024a; Das et al., 2024; Liu et al., 2024; Zhan et al., 2023), our study provides a theoretical foundation for this phenomenon by introducing margin-type conditions to analyze excess classification error in RLHF reward function learning.

**Convergence Analysis for Deep Neural Network Estimators** In this paper, we present the convergence theory based on specific neural network structures, employing uniform convergence analysis (Schmidt-Hieber, 2020; Diakonikolas et al., 2021). The optimization aspects are beyond the scope of this study and we refer to Allen-Zhu et al. (2019); Lyu et al. (2021) for more discussion. Recent studies have proposed other theories regarding the generalization performance of deep neural networks. Some representative studies include norm-based generalization bounds (Bartlett et al., 2017; Ma et al., 2022), uniform stability theory (Arora et al., 2018; Zhang et al., 2017), and algorithm-based generalization bounds (Wang & Ma, 2022). All these studies provide new insights

for understanding the properties of neural network estimators. Some works focus on the convergence theory for Transformer networks. Examples include (Takakura & Suzuki, 2023; Deora et al., 2023; Wang et al., 2024b). In addition, we note that most of these studies focus on regression and classification problems. The analysis for preference learning is relatively limited. Theoretical guarantees for the pairwise comparison model often rely on additional specific conditions related to dataset structures. It is of interest to study the learning theory for the reward modeling with DNN based estimator.

## 5 CONCLUSION

Reward modeling is becoming an essential component of real-world AI systems, crucial for developing large language models and aligning them with human values. Obtaining theoretical guarantees for reward models offers an opportunity to understand the efficiency of RLHF in aligning language models and identifying which values are embedded in them. In this paper, we establish the regret bound of the reward function implemented by a deep neural network in a non-asymptotic framework. Specifically, the stochastic and approximation errors are well-balanced by selecting network configurations that achieve the optimal convergence rate. This fully non-parametric approach effectively addresses reward model misspecification in linear reward settings while being applicable to various pairwise comparison models. Moreover, our sharper bound is based on a margin condition imposed on the comparison dataset rather than directly on the unknown reward function. The condition on the dataset is verifiable with the estimated winning probability while the assumption on the unknown reward function is unverifiable in practice.

We also note that controlling the effects of ambiguous data is important and worthwhile for future research. Prior works developed learning techniques that are compatible with the label noise level in learning halfspaces and active learning (Yan & Zhang, 2017; Zhang et al., 2020). In view of the margin condition, we assert that computationally efficient learning algorithms can also be developed to take advantage of the clear preference structure in human belief. It is also of interest to extend our result to trajectory-based comparisons under Markov decision process settings (Zhu et al., 2023). These considerations can be of additional interest when applied to more general reinforcement learning tasks.

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

## A  TECHNICAL NOTATIONS

**Notations:**  For sequences $\{b_n\}_{n \in \mathbb{N}}$ and $\{c_n\}_{n \in \mathbb{N}}$, we say $b_n = \mathcal{O}(c_n)$ if there exists an absolute constant $k$ and $n_0 \in \mathbb{N}$ such that $b_n \leq k c_n$ for all $n > n_0$. And we say $b_n = \Theta(c_n)$ if there exists absolute constants $k_1, k_2$ and $n_0 \in \mathbb{N}$ such that $k_1 c_n \leq b_n \leq k_2 c_n$ for all $n > n_0$. Let $\lceil u \rceil$ denote the smallest integer that is no less than $u$, and $\lfloor u \rfloor$ denote the greatest integer that is no greater than $u$. For a Lebesgue measurable subset $\mathcal{S} \subseteq \mathbb{R}^d$, by $L^q(\mathcal{S}, \ell^p)$ we denote the function norm for all real-valued functions $f : \mathcal{S} \times \mathcal{A} \to \mathbb{R}$, such that for every $a \in \mathcal{A}$, $f(\cdot, a)$ is Lebesgue measurable on $\mathcal{S}$ and

$$\|f\|_{L^q(\mathcal{S}, \ell^p)} := \begin{cases} \left( \int_{\mathcal{S}} \left( \sum_{a \in \mathcal{A}} |f(s, a)|^p \right)^{\frac{q}{p}} d\rho_s \right)^{\frac{1}{q}}, & 1 \leq q < +\infty, \\ \operatorname{ess\,sup}_{s \in \mathcal{S}} \left( \sum_{a \in \mathcal{A}} |f(s, a)|^p \right)^{\frac{1}{p}}, & q = +\infty \end{cases} \tag{6}$$

is finite.

**Covering Number of a Function Class:**  Let $\mathcal{F}$ be a class of functions $: \mathcal{X} \to \mathbb{R}$. For a given $\epsilon > 0$, we denote $\mathcal{N}(\mathcal{F}, \epsilon, \|\cdot\|)$ as the covering number of $\mathcal{F}$ with radius $\delta$ under some norm $\|\cdot\|$ as the least cardinality of a subset $\mathcal{F}' \subseteq \mathcal{F}$, satisfying

$$\sup_{f \in \mathcal{F}} \min_{f' \in \mathcal{F}'} \|f - f'\| \leq \delta.$$

This quantity measures the minimum number of functions in $\mathcal{F}$ needed to cover the set of functions within a distance of $\delta$ under the norm $\|\cdot\|$.

**Laplacian Matrix and Graph Structure:**  The connectivity of the comparison graph plays a crucial role in estimating from pairwise data (Mohar et al., 1991), which relates to the second smallest eigenvalue of $\Lambda$, we denote it as $\lambda_2(\Lambda) := \min_{w \perp \mathbf{1}} w^\top \Lambda w / w^\top w$. The pair of actions being compared need to be carefully selected for effective reward modeling, ensuring that $\lambda_2(\Lambda)$ is not excessively small. If some actions are almost not queried among the data, the spectral gap $1/\lambda_2(\Lambda)$ diverges and thus makes the MLE estimator degenerate in terms of regret. This happens if the number of comparisons between some pairs is too scarce, as shown in Example 3, where $1/\lambda_2(L)$ can blow up to an order of $\Theta(n)$, so there is no guarantee for the regret bounded.

**Example 3** *Suppose there are four actions in $\mathcal{A}$. Let $n = 2t + 1$, we query $(a_1, a_2), (a_2, a_3)$ for $t$ times each, and $(a_3, a_4)$ only once. The Laplacian matrix of this pairwise comparison design is*

$$\Lambda = \frac{1}{n} \begin{bmatrix} t & -t & 0 & 0 \\ -t & 2t & -t & 0 \\ 0 & -t & t+1 & -1 \\ 0 & 0 & -1 & 1 \end{bmatrix}$$

*It is clear that $1/\lambda_2(\Lambda) \asymp 3(2t+1)/2 = \Theta(n)$, which blows up the error $\|\hat{r} - r^*\|^2_{L^2(\mathcal{S}, \ell^2)}$, although the excess risk is still under control.*

It is worth pointing out that the optimal choice of $\Lambda$ should satisfy that $\lambda_2(\Lambda) = \Theta(1/|\mathcal{A}|)$. As $\Lambda$ is the Laplacian matrix whose $\operatorname{trace}(\Lambda) = 2$, $\lambda_2(\Lambda) \leq 2/(|\mathcal{A}| - 1)$. A natural choice to reach this bound is to require every pair of actions to be compared equally. In the terminology of the graph, this choice is referred to as the *complete graph*. In addition to the *complete graph*, there are several graphs satisfying $\lambda_2(\Lambda) = \Theta(1/|\mathcal{A}|)$, such as the *complete bipartite* graph and *star* graph. In contrast, the *path* graph and *cycle* graph lead to $\lambda_2(\Lambda) = \Theta(1/|\mathcal{A}|^3)$ (Shah et al., 2016). Therefore, effective reward modeling requires careful selection of the pairwise comparison subset.

## B  PROOF OF LEMMA 1

We define

$$G(u) := \int_0^\infty g(y, u) dy - \frac{1}{2}.$$

By definition of the comparison function $g$, it is straightforward that $G(0) \leq 0$, so we can always find a tangent line to $G(u)$ crossing the origin on $\mathbb{R}^+$, *i.e.,* there exists a constant $\kappa \leq \sup_{|u| \leq c_{r^*}} |\frac{d}{du} G(u)|$, such that $G(u) \leq \kappa u$. Note that $G(u/\kappa) \leq u$ also holds for all $u \in \mathbb{R}^+$. Then the set

$$\left\{ s \in \mathcal{S} : G(r^*(s, \pi_{r^*}(s)) - r^*(s, a')) \leq G\left(\frac{t}{\kappa}\right) \right\} \subseteq \left\{ s \in \mathcal{S} : G(r^*(s, \pi_{r^*}(s)) - r^*(s, a')) \leq t \right\}.$$

Also, due to the monotonicity of the function $G(u)$, the set

$$\left\{ s \in \mathcal{S} : G(r^*(s, \pi_{r^*}(s)) - r^*(s, a')) \leq G\left(\frac{t}{\kappa}\right) \right\} = \left\{ s \in \mathcal{S} : r^*(s, \pi_{r^*}(s)) - r^*(s, a') \leq \frac{t}{\kappa} \right\}.$$

Thus,

$$\int_{\mathcal{S}} \mathbf{1} \left\{ r^*(s, \pi_{r^*}(s)) - r^*(s, a') \leq \frac{t}{\kappa} \right\} d\rho_s \leq c_g t^{\frac{\alpha}{1-\alpha}}.$$

Replace $t/\kappa$ with $t$ and update the constant results in the desired inequality.

**Example 4** *Here we consider the BT model as an example (see Example 1). We deliberately set $t' = 2t$, where $t$ is the probability gap in Assumption 1.*

$$\int_{\mathcal{S}} \mathbf{1} \left\{ r^*(s, r^*(s, \pi_{r^*}(s))) - r^*(s, a') \leq t' \right\} d\rho_s$$

$$= \int_{\mathcal{S}} \mathbf{1} \left\{ \frac{\exp(r^*(s, \pi_{r^*}(s)))}{\exp(r^*(s, \pi_{r^*}(s))) + \exp(r^(s, a'))} - \frac{1}{2} \leq \frac{1}{2} \frac{e^{t'} - 1}{e^{t'} + 1} \right\} d\rho_s.$$

$$= \mathbb{P}_{\mathcal{S}} \left( \mathbb{P}(y > 0 \mid s, a_1 = \pi_{r^*}(s), a_0 = a') - 1/2 \leq \frac{1}{2} \frac{e^{t'} - 1}{e^{t'} + 1} \right)$$

$$\leq \mathbb{P}_{\mathcal{S}} \left( \mathbb{P}(y > 0 \mid s, a_1 = \pi_{r^*}(s), a_0 = a') - 1/2 \leq t'/4 \right)$$

$$\leq (1/4)^{\frac{\alpha}{1-\alpha}} c_g t'^{\frac{\alpha}{1-\alpha}}.$$

*The first step holds since $\frac{1}{2} \frac{e^{t'} - 1}{e^{t'} + 1}$ is monotonically increasing. The third is due to $\frac{t'}{4} \geq \frac{1}{2} \frac{e^{t'} - 1}{e^{t'} + 1}$ for all $t' \in (0, 1)$. Replacing the notation $t'$ with $t$ yields the desired inequality.*

**Example 5** *In the Thurstonian model (see Example 2), we set $t' = \sqrt{\pi/2}\, t$, where $t$ is the probability gap in Assumption 1.*

$$\int_{\mathcal{S}} \mathbf{1} \left\{ r^*(s, r^*(s, \pi_{r^*}(s))) - r^*(s, a') \leq t' \right\} d\rho_s$$

$$= \int_{\mathcal{S}} \mathbf{1} \left\{ \frac{\exp(-(r^*(s, r^*(s, \pi_{r^*}(s))) - r^*(s, a'))^2/2)}{\sqrt{2\pi}} - \frac{1}{2} \leq \frac{e^{-t'^2/2}}{\sqrt{2\pi}} - \frac{1}{2} \right\} d\rho_s.$$

$$= \mathbb{P}_{\mathcal{S}} \left( \mathbb{P}(y > 0 \mid s, a_1 = \pi_{r^*}(s), a_0 = a') - 1/2 \leq \frac{e^{-t'^2/2}}{\sqrt{2\pi}} - \frac{1}{2} \right)$$

$$\leq \mathbb{P}_{\mathcal{S}} \left( \mathbb{P}(y > 0 \mid s, a_1 = \pi_{r^*}(s), a_0 = a') - 1/2 \leq t'/\sqrt{2\pi} \right)$$

$$\leq (1/\sqrt{2\pi})^{\frac{\alpha}{1-\alpha}} c_g t'^{\frac{\alpha}{1-\alpha}}.$$

*The first step holds since $\frac{e^{-t'^2/2}}{\sqrt{2\pi}} - \frac{1}{2}$ is monotonically increasing. The third is due to $t'/\sqrt{2\pi} \geq \frac{e^{-t'^2/2}}{\sqrt{2\pi}} - \frac{1}{2}$ for all $t' \in (0, 1)$. Replacing the notation $t'$ with $t$ yields the desired inequality.*

## C PROOF OF THEOREM 1

By definition of the regret,

$$\mathcal{E}(r) = \int_{\mathcal{S}} \left( r^*(s, \pi_{r^*}(s)) - r^*(s, \pi_{r^*}(s)) \right) d\rho_s$$

$$= \int_{\pi_{r^*}(s) \neq \pi_{\hat{r}}(s)} \left( r^*(s, \pi_{r^*}(s)) - r^*(s, \pi_{\hat{r}}(s)) \right) d\rho_s.$$

Now, we define two sets, given any $\eta \in (0, 1)$,

$$\mathcal{S}_1 = \{ s \in \mathcal{S} : 0 < r^*(s, \pi_{r^*}(s)) - \max_{a \in \mathcal{A}/\pi_{r^*}(s)} r_a^*(s) \leq \eta \};$$

$$\mathcal{S}_2 = \{ s \in \mathcal{S} : r^*(s, \pi_{r^*}(s)) - \max_{a \in \mathcal{A}/\pi_{r^*}(s)} r_a^*(s) > \eta \}.$$

It is worth noting that the two sets $\mathcal{S}_1$ and $\mathcal{S}_2$ are the complement of each other, and they are independent of any reward estimator $r$. Then, we can decompose the performance loss as follows,

$$\mathcal{E}(r) = \int_{\pi_{r^*}(s) \neq \pi_r(s) \cap \mathcal{S}_1} \left( r^*(s, \pi_{r^*}(s)) - r^*(s, \pi_r(s)) \right) d\rho_s$$

$$+ \int_{\pi_{r^*}(s) \neq \pi_r(s) \cap \mathcal{S}_2} \left( r^*(s, \pi_{r^*}(s)) - r^*(s, \pi_r(s)) \right) d\rho_s$$

$$= \int_{\pi_{r^*}(s) \neq \pi_r(s) \cap \mathcal{S}_1} \left( r^*(s, \pi_{r^*}(s)) - r^*(s, \pi_r(s)) \right) d\rho_s$$

$$+ \int_{\pi_{r^*}(s) \neq \pi_r(s) \cap \mathcal{S}_2} \left( r^*(s, \pi_{r^*}(s)) - r^*(s, \pi_r(s)) \right) \mathbf{1} \left\{ \sum_{a \in \mathcal{A}} |r(s, a) - r^*(s, a)| \leq \eta \right\} d\rho_s$$

$$+ \int_{\pi_{r^*}(s) \neq \pi_r(s) \cap \mathcal{S}_2} \left( r^*(s, \pi_{r^*}(s)) - r^*(s, \pi_r(s)) \right) \mathbf{1} \left\{ \sum_{a \in \mathcal{A}} |r(s, a) - r^*(s, a)| \geq \eta \right\} d\rho_s.$$

$$(7)$$

Note that the second term in the last step is $0$ since there is no regret loss as long as the estimated action is the optimal action.

$$\mathcal{E}(r) \leq \eta \int_{\mathcal{S}} \mathbf{1} \left\{ 0 < r^*(s, \pi_{r^*}(s)) - \max_{a \in \mathcal{A}/\pi_{r^*}(s)} r_a^*(s) \leq \eta \right\} d\rho_s$$

$$+ 0 + c_{r^*} \int_{\pi_{r^*}(s) \neq \pi_r(s) \cap \mathcal{S}_2} \mathbf{1} \left\{ \sum_{a \in \mathcal{A}} |r(s, a) - r^*(s, a)| \geq \eta \right\} d\rho_s.$$

Recall that with Lemma 1, for a non-negative random variable $X$, and non-decreasing function $\varphi(u) > 0$, the Markov inequality states $\varphi(u)\mathbb{P}(X \geq u) \leq \mathbb{E}(\varphi(X))$. Let $\varphi(\eta) = \eta^2$, where $\alpha \in (0, 1)$. Then,

$$\eta^2 \int_{\pi_{r^*}(s) \neq \pi_r(s) \cap \mathcal{S}_2} \mathbf{1} \left\{ \sum_{a \in \mathcal{A}} |r(s, a) - r^*(s, a)| \geq \eta \right\} d\rho_s$$

$$\leq \eta^2 \int_{\mathcal{S}} \mathbf{1} \left\{ \sum_{a \in \mathcal{A}} |r(s, a) - r^*(s, a)| \geq \eta \right\} d\rho_s \leq \int_{\mathcal{S}} \left( \sum_{a \in \mathcal{A}} |r(s, a) - r^*(s, a)| \right)^2 d\rho_s,$$

where we apply the Markov inequality in the last step, taking an expectation over the state space $\mathcal{S}$. Note that $\|r - r^*\|_{L^2(\mathcal{S}, \ell^1)}^2 \leq |\mathcal{A}| \|r - r^*\|_{L^2(\mathcal{S}, \ell^2)}^2$, then

$$\mathcal{E}(r) \leq \eta \cdot c_g \eta^{\frac{\alpha}{1-\alpha}} + (\frac{1}{\eta})^2 |\mathcal{A}| c_{r^*} \|r - r^*\|_{L^2(\mathcal{S}, \ell^2)}^2.$$

To balance the two terms above, we choose

$$\eta = \left( \frac{|\mathcal{A}| c_{r^*}}{c_g} \|r - r^*\|_{L^2(\mathcal{S}, \ell^2)}^2 \right)^{\frac{1-\alpha}{3-2\alpha}}.$$

Consequently, we have

$$\mathcal{E}(r) \leq c_1 \left( \|r - r^*\|^2_{L^2(\mathcal{S}, \ell^2)} \right)^{\frac{1}{3-2\alpha}}$$

with $c_1 = \left( \frac{|\mathcal{A}| c_{r^*}}{c_g} \right)^{\frac{1-\alpha}{3-2\alpha}}$.

$\square$

## C.1 PROOF OF COROLLARY 1

The proof generally follows the proof of Theorem 1 but without any margin condition on the state distribution. Starting from the second step of (7), we have

$$\mathcal{E}(r) = \int_{\pi_{r^*}(s) \neq \pi_r(s) \cap \mathcal{S}_1} \left( r^*(s, \pi_{r^*}(s)) - r^*(s, \pi_r(s)) \right) d\rho_s$$

$$+ \int_{\pi_{r^*}(s) \neq \pi_r(s) \cap \mathcal{S}_2} \left( r^*(s, \pi_{r^*}(s)) - r^*(s, \pi_r(s)) \right) d\rho_s$$

$$= \int_{\pi_{r^*}(s) \neq \pi_r(s) \cap \mathcal{S}_1} \left( r^*(s, \pi_{r^*}(s)) - r^*(s, \pi_r(s)) \right) d\rho_s$$

$$+ \int_{\pi_{r^*}(s) \neq \pi_r(s) \cap \mathcal{S}_2} \left( r^*(s, \pi_{r^*}(s)) - r^*(s, \pi_r(s)) \right) \mathbf{1} \left\{ \sum_{a \in \mathcal{A}} |r(s, a) - r^*(s, a)| \leq \eta \right\} d\rho_s$$

$$+ \int_{\pi_{r^*}(s) \neq \pi_r(s) \cap \mathcal{S}_2} \left( r^*(s, \pi_{r^*}(s)) - r^*(s, \pi_r(s)) \right) \mathbf{1} \left\{ \sum_{a \in \mathcal{A}} |r(s, a) - r^*(s, a)| \geq \eta \right\} d\rho_s$$

$$\leq \eta + 0 + c_{r^*} \int_{\pi_{r^*}(s) \neq \pi_r(s) \cap \mathcal{S}_2} \mathbf{1} \left\{ \sum_{a \in \mathcal{A}} |r(s, a) - r^*(s, a)| \geq \eta \right\} d\rho_s$$

$$\leq \eta + (\frac{1}{\eta})^2 |\mathcal{A}| c_{r^*} \|r - r^*\|^2_{L^2(\mathcal{S}, \ell^1)}.$$

Note that $\|r - r^*\|^2_{L^2(\mathcal{S}, \ell^1)} \leq |\mathcal{A}| \|r - r^*\|^2_{L^2(\mathcal{S}, \ell^2)}$. Then with $\eta = (|\mathcal{A}| c_{r^*} \|r - r^*\|^2_{L^2(\mathcal{S}, \ell^2)})^{1/3}$ and $c_2 = (|\mathcal{A}| c_{r^*})^{1/3}$, we have

$$\mathcal{E}(C_r) \leq \eta + (\frac{1}{\eta})^2 |\mathcal{A}| c_{r^*} \|r - r^*\|^2_{L^2(\mathcal{S}, \ell^2)} \leq c_2 \left( \|r - r^*\|^2_{L^2(\mathcal{S}, \ell^2)} \right)^{\frac{1}{3}}.$$

$\square$

## C.2 DISCUSSION ON SELECTION CONSISTENCY

In the main text, we present the regret bound in order of $\mathcal{O}((\|r - r^*\|^2_{L^2(\mathcal{S}, \ell^2)})^{1/(3-2\alpha)})$. It is often of interest to give the result of the selection consistency for a given reward estimator $r$.

**Lemma 3** *Given the Assumption 1 and the estimator from (3), there exist an universal constant $c_6$ such that*

$$\mathbb{P}_{\mathcal{S}} \left( \pi_{r^*}(s) \neq \pi_r(s) \right) \leq c_6 \left( \|r - r^*\|^2_{L^2(\mathcal{S}, \ell^2)} \right)^{\frac{\alpha}{3-2\alpha}},$$

*where $\pi_{r^*}(s)$ is an optimal policy that gives the optimal action that maximizes the reward.*

This quantity is important in statistical machine learning literature (Peter L Bartlett & McAuliffe, 2006; Audibert & Tsybakov, 2007). As $\alpha \to 1$, the selection consistency achieves the best rate. When $\alpha \to 0$, following the Lemma 1, the reward of an alternative action is comparable with the optimal one; thus, the selection consistency result may fail. Fundamentally, in reinforcement learning problems, we do not make predictions that are consistent with data labels. Still, it is a beneficial complement for us to understand the effects of the margin-type condition.

## C.3 PROOF OF LEMMA 3

By definition of the regret,

$$
\begin{aligned}
\mathcal{E}(r) &= \int_{\mathcal{S}} \left( r^*(s, \pi_{r^*}(s)) - r^*(s, \pi_r(s)) \right) d\rho_s \\
&\geq \int_{\mathcal{S}} \left( r^*(s, \pi_{r^*}(s)) - r^*(s, \pi_r(s)) \right) \mathbf{1} \left\{ r^*(s, \pi_{r^*}(s)) - \max_{a \in \mathcal{A}/\pi_{r^*}(s)} r_a^*(s) > t \right\} d\rho_s \\
&\geq t \int_{\mathcal{S}} \mathbf{1} \left\{ \pi_{r^*}(s) \neq \pi_r(s) \right\} \cdot \mathbf{1} \left\{ r^*(s, \pi_{r^*}(s)) - \max_{a \in \mathcal{A}/\pi_{r^*}(s)} r_a^*(s) > t \right\} d\rho_s \\
&\geq t \left( \mathbb{P}_{\mathcal{S}} \left( \pi_{r^*}(s) \neq \pi_r(s) \right) - c_g t^{\frac{\alpha}{1-\alpha}} \right).
\end{aligned}
$$

The last line follows $\mathbb{P}(E_1 \cap E_2) \geq \mathbb{P}(E_1) - \mathbb{P}(E_2^c)$, for any two events $E_1$ and $E_2$. Minimizing this term with respect to $t$, *i.e.*, taking $t = c' \mathbb{P}_{\mathcal{S}} \left( \pi_{r^*}(s) \neq \pi_r(s) \right)^{(1-\alpha)/\alpha}$ results in

$$
\mathbb{P}_{\mathcal{S}} \left( \pi_{r^*}(s) \neq \pi_r(s) \right) \leq \frac{1}{c'^\alpha} \mathcal{E}(r)^\alpha \leq c_6 \left( \| r - r^* \|_{L^2(\mathcal{S}, \ell^2)}^2 \right)^{\frac{\alpha}{3-2\alpha}},
$$

where $c'$ and $c_6$ are constants depending on $c_g$ and $\alpha$. $\square$

# D PROOF OF ERROR BOUNDS

## D.1 PROOF OF LEMMA 2

Denote $\tilde{r}$ as an estimator that maximizes the likelihood in the function class $\mathcal{F}_{\text{DNN}}$ as

$$
\tilde{r} = \arg\max_{r \in \mathcal{F}_{\text{DNN}}} l(r).
$$

We expand the excess risk by adding and substituting the following terms:

$$
\begin{aligned}
l(r^*) - l(\hat{r}) &= \left[ \hat{l}(\hat{r}) - l(\hat{r}) \right] + \left[ \hat{l}(\tilde{r}) - \hat{l}(\hat{r}) \right] + \left[ l(\tilde{r}) - \hat{l}(\tilde{r}) \right] + \left[ l(r^*) - l(\tilde{r}) \right] \\
&\leq \left[ l(\hat{r}) - \hat{l}(\hat{r}) \right] + \left[ l(\tilde{r}) - \hat{l}(\tilde{r}) \right] + \left[ l(r^*) - l(\tilde{r}) \right] \\
&\leq 2 \sup_{r \in \mathcal{F}_{\text{DNN}}} \left| l(r) - \hat{l}(r) \right| + l(r^*) - l(\tilde{r}) \\
&= 2 \sup_{r \in \mathcal{F}_{\text{DNN}}} \left| l(r) - \hat{l}(r) \right| + \inf_{r \in \mathcal{F}_{\text{DNN}}} \left[ l(r^*) - l(r) \right],
\end{aligned}
$$

where the first inequality follows from the definition of $\hat{r}$ as the maximizer of $\hat{l}(r)$ in $\mathcal{F}_{\text{DNN}}$, then $\hat{l}(\tilde{r}) - \hat{l}(\hat{r}) \leq 0$. The second inequality holds due to the fact that both $\hat{r}$ and $\tilde{r}$ belong to the function class $\mathcal{F}_{\text{DNN}}$, and the last equality is valid by the definition of $\tilde{r}$.

## D.2 PROOF OF PROPOSITION 1

Let $Z_i(r) := \log g \left( y^i; r(s^i, a_1^i) - r(s^i, a_0^i) \right)$ for $1 \leq i \leq N$. Then, given Assumption 2 and Hoeffding's inequality, with probability at least $1 - \delta$, we have

$$
\left| \frac{1}{N} \sum_{i=1}^{N} \left( Z_i(r) - \mathbb{E}\left[ Z_i(r) \right] \right) \right| \leq \kappa_0 \sqrt{\frac{\log\left(\frac{2}{\delta}\right)}{2N}},
$$

where the expectation $\mathbb{E}$ is taken over the data distribution. Now considering an estimator $r$ that is within the deep neural network function class, We can further obtain that for any given $\tau > 0$, let $r_1, r_2, \ldots, r_{\mathcal{N}}$ be the anchor points of an $\tau$-covering for the function class $\mathcal{F}_{\text{DNN}}$, where we denote $\mathcal{N} := \mathcal{N}\left( \mathcal{F}_{\text{DNN}}, \tau, \| \cdot \|_{L^\infty(\mathcal{S}, \ell^\infty)} \right)$ as the covering number of $\mathcal{F}_{\text{DNN}}$ with radius $\tau$ under the norm

$\| \cdot \|_{L^\infty(\mathcal{S}, \ell^\infty)}$. By definition, for any $r \in \mathcal{F}_{\text{DNN}}$, there exists an anchor $r_h$ for $h \in \{1, \ldots, \mathcal{N}\}$ such that $\|r_h - r\|_{L^\infty(\mathcal{S}, \ell^\infty)} \leq \tau$. We further decompose the stochastic error as follows

$$
\begin{aligned}
l(r) - \hat{l}(r) \leq & l(r_h) - \hat{l}(r_h) + l(r) - l(r_h) + \hat{l}(r_h) - \hat{l}(r) \\
= & l(r_h) - \hat{l}(r_h) + \frac{1}{N}\left|\sum_{i=1}^N \mathbb{E}[Z_i(r) - Z_i(r_h)]\right| + \frac{1}{N}\left|\sum_{i=1}^N [Z_i(r) - Z_i(r_h)]\right| \\
\leq & l(r_h) - \hat{l}(r_h) + 2\kappa_1 \tau,
\end{aligned}
$$

where the last step is due to the Lipschitz property of $\log g(y, u)$. Therefore, with a fixed $\epsilon > 0$,

$$
\begin{aligned}
\mathbb{P}\left(\sup_{r \in \mathcal{F}_{\text{DNN}}} \left|l(r) - \hat{l}(r)\right| \geq (\epsilon + \kappa_1 \tau)\right) & \leq \mathbb{P}\left(\exists\, h \in \{1, \ldots, \mathcal{N}\} : \left|l(r_h) - \hat{l}(r_h)\right| \geq \epsilon\right) \\
& \leq \mathcal{N}_n\left(\mathcal{F}_{\text{DNN}}, \tau, \| \cdot \|_{L^\infty(\mathcal{S}, \ell^\infty)}\right) \max_{h \in \{1, \ldots, \mathcal{N}\}} \mathbb{P}\left(\left|l(r_h) - \hat{l}(r_h)\right| \geq \epsilon\right) \quad (8) \\
& \leq 2\mathcal{N}_n\left(\mathcal{F}_{\text{DNN}}, \tau, \| \cdot \|_{L^\infty(\mathcal{S}, \ell^\infty)}\right) \exp\left(-\frac{2N\epsilon^2}{\kappa_0^2}\right),
\end{aligned}
$$

where the last line comes from Hoeffding's inequality. Then, for any $\delta > 0$, let $\tau = 1/N$ and $\epsilon = \kappa_0\sqrt{2\log\left(2\mathcal{N}\left(\mathcal{F}_{\text{DNN}}, \tau, \| \cdot \|_{L^\infty(\mathcal{S}, \ell^\infty)}\right)/\delta\right)/N}$ so that the right-hand side of (8) equals to $\delta$, we have

$$
\mathbb{P}\left(\sup_{r \in \mathcal{F}_{\text{DNN}}} \left|l(r) - \hat{l}(r)\right| \geq \epsilon + 2\kappa_1 \tau\right) \leq \delta.
$$

In other words, with probability at least $1 - \delta$,

$$
\begin{aligned}
\sup_{r \in \mathcal{F}_{\text{DNN}}} \left|l(r) - \hat{l}(r)\right| & \leq \epsilon + 2\kappa_1 \tau \\
& \leq \sqrt{2}\kappa_0\left(\sqrt{\frac{\log 2\sqrt{2}\mathcal{N}\left(\tau, \mathcal{F}_{\text{DNN}}, \| \cdot \|_{L^\infty(\mathcal{S}, \ell^\infty)}\right)}{N}} + \sqrt{\frac{\log\left(\frac{1}{\delta}\right)}{N}}\right) + \frac{2\kappa_1}{N},
\end{aligned} \quad (9)
$$

where in the second step we use the inequality $\sqrt{C_1 + C_2} \leq \sqrt{C_1} + \sqrt{C_2}$, for any $C_1, C_2 \geq 0$.

In the rest of the proof, we bound the covering number. Without loss of generality, we also define classes of sub-networks of $\mathcal{F}_{\text{DNN}}$, that is, $\{\mathcal{F}_1, \mathcal{F}_2, \cdots, \mathcal{F}_{|\mathcal{A}|}\}$, with non-sharing hidden layers. By doing so, the function $r_a \in \mathcal{F}_a$ in each reduced function class takes states as input and returns $r(s, a)$ given an action $a \in \mathcal{A}$. For convenience, we assume all the sub-networks have the same width in each layer, and all model parameters are bounded within $[-1, 1]$ that is,

$$
\mathcal{F}_a(W, D) = \left\{r_a(\cdot; \theta) : \mathbb{R}^d \to \mathbb{R} \text{ defined in (4)} : \theta \in [-1, 1]^p\right\},
$$

such that the function class of our interest is covered by the product space of sub-network classes, that is, $\mathcal{F}_{\text{DNN}} \subset \mathcal{F}_1 \otimes \mathcal{F}_2 \otimes \cdots \otimes \mathcal{F}_{|\mathcal{A}|}$. Now, we can express the covering number in terms of the product of complexities of sub-networks by

$$
\log \mathcal{N}\left(\mathcal{F}_{\text{DNN}}, \tau, \| \cdot \|_{L^\infty(\mathcal{S}, \ell^\infty)}\right) \leq |\mathcal{A}| \log \mathcal{N}(\mathcal{F}_a, \tau/|\mathcal{A}|, \| \cdot \|_{L^\infty}). \quad (10)
$$

For the deep ReLU neural network in our setting, Shen (2024, Theorem 2) shows that for any $\tau > 0$.

$$
\mathcal{N}(\mathcal{F}, \tau, \| \cdot \|_{\mathcal{L}_\infty}) \leq \frac{\left(2^{D+5}(D+1)W^D \cdot \tau^{-1}\right)^p}{(W!)^D}.
$$

Then, apply the above inequality to (10),

$$
\log \mathcal{N}\left(\mathcal{F}_{\text{DNN}}, \tau, \| \cdot \|_{L^\infty(\mathcal{S}, \ell^\infty)}\right) \leq |\mathcal{A}| Dp \log\left(\frac{2W(32(D+1)|\mathcal{A}|/\tau)^{1/D}}{(W!)^{1/p}}\right). \quad (11)
$$

Plug (11) in (9), and take $\tau = 1/N$, with probability at least $1 - \delta$,

$$
\sup_{r \in \mathcal{F}_{\text{DNN}}} \left| l(r) - \hat{l}(r) \right|
$$

$$
\leq \kappa_0 \sqrt{\frac{2}{N}} \left( \sqrt{\log 2\sqrt{2} |\mathcal{A}| Dp \log \left( \frac{2W(32(D+1)|\mathcal{A}|N)^{1/D}}{(W!)^{1/p}} \right)} + \sqrt{\log(1/\delta)} \right) + \frac{2\kappa_1}{N}
$$

$$
\leq \kappa_0 \sqrt{\frac{2}{N}} \left( c_3 \sqrt{|\mathcal{A}| Dp \log \left( \frac{W((D+1)|\mathcal{A}|N)^{1/D}}{(W!)^{1/p}} \right)} + \sqrt{\log(1/\delta)} \right).
$$

In the second step, due to that $1/N$ is decaying faster than $\sqrt{\log(N)/N}$, we can simplify the expression by making $c_3$ a universal constant. $\square$

## D.3 PROOF OF PROPOSITION 2

We adopt the ReLU network approximation result for Hölder smooth functions in $\mathcal{H}^\beta([0,1]^d, c_{\mathcal{H}})$, proposed in Jiao et al. (2023, Theorem 3.3). With Assumption 3, for any $M_1, M_2 \in \mathbb{N}^+$, and for each $a \in \mathcal{A}$, there exists a function $\tilde{r}$ implemented by a ReLU network with width $W = 38(\lfloor \beta \rfloor + 1)^2 d^{\lfloor \beta \rfloor + 1} M_1 \lceil \log_2(8M_1) \rceil$ and depth $D = 21(\lfloor \beta \rfloor + 1)^2 M_2 \lceil \log_2(8M_2) \rceil$ such that

$$
|\tilde{r}(s, a) - r^*(s, a)| \leq 18 c_{\mathcal{H}} (\lfloor \beta \rfloor + 1)^2 d^{\lfloor \beta \rfloor + (\beta \vee 1)/2} (M_1 M_2)^{-2\beta/d},
$$

for all $s \in [0,1]^d$ except a small set $\Omega \in \mathcal{S}$ with Lebesgue measure $\delta K p$, where $\delta$ can be arbitrarily small. Since we have the same setting for all sub-networks, we have the same bound for $\max_{a \in \mathcal{A}} |\tilde{r}(s, a) - r^*(s, a)|$. Therefore, integrate both sides with respect to the state space distribution we have

$$
\|\tilde{r} - r^*\|_{L^1(\mathcal{S}, \ell^\infty)} \leq 18 c_{\mathcal{H}} (\lfloor \beta \rfloor + 1)^2 p^{\lfloor \beta \rfloor + (\beta \vee 1)/2} (M_1 M_2)^{-2\beta/d}
$$
$$
+ \mathbb{P}(\Omega) \cdot \sup_{s \in \Omega} \{ \max_{a \in \mathcal{A}} |\tilde{r}(s, a) - r^*(s, a)| \}. \quad (12)
$$

By Assumption 3, the marginal distribution of $\mathcal{S}$ is absolutely continuous with respect to the Lebesgue measure, which means that $\liminf_{\delta \to 0} \mathbb{P}(\Omega) = 0$. Meanwhile, we know from the definition of $\mathcal{F}_{\text{DNN}}$ and Assumption 2 that both $\sup_{s \in \mathcal{S}} |\tilde{r}(s, a)|$ and $\sup_{s \in \mathcal{S}} |r^*(s, a)|$ are bounded for all $a \in \mathcal{A}$. Therefore, by taking the limit infimum with respect to $\delta$ on both sides of (12), we have

$$
\|\tilde{r} - r^*\|_{L^1(\mathcal{S}, \ell^\infty)} \leq 18 c_{\mathcal{H}} (\lfloor \beta \rfloor + 1)^2 d^{\lfloor \beta \rfloor + (\beta \vee 1)/2} (M_1 M_2)^{-2\beta/d}.
$$

Therefore, by the Lipschitz property of the likelihood function,

$$
l(r^*) - l(\tilde{r}) \leq 2\kappa_1 \|\tilde{r} - r^*\|_{L^1(\mathcal{S}, \ell^\infty)} \leq 36 \kappa_1 c_{\mathcal{H}} (\lfloor \beta \rfloor + 1)^2 d^{\lfloor \beta \rfloor + (\beta \vee 1)/2} (M_1 M_2)^{-2\beta/d}.
$$

$\square$

## D.4 PROOF OF THEOREM 3

Before dealing with the error terms, we first control the logarithmic term involved in the covering number. Let $c$ be a positive universal constant, then

$$
\frac{W((D+1)|\mathcal{A}|N)^{1/D}}{(W!)^{1/p}} \leq \frac{W((D+1)|\mathcal{A}|N)^{1/D}}{\left( e \left( W/e \right)^W \right)^{1/p}}
$$
$$
= c \left( \frac{W((D+1)|\mathcal{A}|N)^{1/D}}{W^{1/\mathcal{W}\mathcal{D}}} \right)
$$
$$
= c \left( W(|\mathcal{A}|N)^{1/D} \right).
$$

The first step is from the inequality that $n! \geq e(n/e)^n$, for any $n \in \mathbb{N}^+$. The second step holds for rectangular neural networks, where the size $p = \mathcal{O}(W^2 D)$. The last step follows Proposition 2,

since we require the network has width $W > 114$ and depth $D > 63$, leading that $W^{1/W} \asymp 1$ and $(D+1)^{1/D} \asymp 1$. Now, we combine both stochastic error and approximation error.

$$l(\hat{r}) - l(r^*) \le 2\kappa_0 \sqrt{\frac{2}{N}} \left( c_3 \sqrt{|\mathcal{A}| D p \log\left(c\left(W(|\mathcal{A}|N)^{1/D}\right)\right)} + \sqrt{\log(1/\delta)} \right)$$
$$+ 36\kappa_1 c_{\mathcal{H}}(\lfloor\beta\rfloor + 1)^2 d^{\lfloor\beta\rfloor + (\beta\vee 1)/2}(M_1 M_2)^{-2\beta/d}.$$

To balance these two error terms, proper tuning parameters $M_1$ and $M_2$ are selected to optimize the convergence rate with the most efficient network design. Since the network size grows linearly in-depth $D$ but quadratically in width $W$. To reach the optimal rate while, at the same time, saving network size, we decide to fix $M_1 = 1$ so that the width $W$ is growing with a polynomial of the input dimension $d$ while independent of sample size $N$. Meanwhile, we take $M_2 = \lfloor N^{d/(2d+4\beta)} \rfloor$. Then with $d \gg \beta$, we have

$$W = 114(\lfloor\beta\rfloor + 1)^2 d^{\lfloor\beta\rfloor + 1}; D = 21(\lfloor\beta\rfloor + 1)^2 \left\lceil N^{\frac{d}{2d+4\beta}} \log_2\left(8N^{\frac{d}{2d+4\beta}}\right) \right\rceil = \mathcal{O}\left(\sqrt{N}\right), \quad (13)$$

and

$$p = \mathcal{O}\left(W^2 D\right) = \mathcal{O}\left((\lfloor\beta\rfloor + 1)^6 d^{2\lfloor\beta\rfloor + 2} \left\lceil N^{\frac{d}{2(d+2\beta)}} (\log_2 N) \right\rceil\right) = \mathcal{O}\left(\sqrt{N}\right),$$

where the $\log(N)$ factors are omitted for simplicity.

Therefore, combining (13) with the error bound, and letting $c_3'$ an another universal constant, with probability at least $1 - \delta$,

$$l(\hat{r}) - l(r^*) \le 2\kappa_0 \sqrt{\frac{2}{N}} \left( c_3' \sqrt{|\mathcal{A}|}(\lfloor\beta\rfloor + 1)^4 d^{\lfloor\beta\rfloor + 1}(\log(N))^2 N^{\frac{d}{2d+4\beta}} + \sqrt{\log(1/\delta)} \right)$$
$$+ 36\kappa_1 c_{\mathcal{H}}(\lfloor\beta\rfloor + 1)^2 d^{\lfloor\beta\rfloor + (\beta\vee 1)/2} N^{\frac{-\beta}{d+2\beta}}$$
$$\le 2\sqrt{2}\kappa_0 \left( c_4 \sqrt{|\mathcal{A}|}(\lfloor\beta\rfloor + 1)^4 d^{\lfloor\beta\rfloor + 1}(\log(N))^2 N^{\frac{-\beta}{d+2\beta}} + \sqrt{\frac{\log(1/\delta)}{N}} \right),$$

where Since the stochastic error and approximation error are of the same order now, in the last step, we combine them with $c_4$, another universal constant. In this part, we find the following connection between the excess risk $l(\hat{r}) - l(r^*)$ and the estimation error $\|\hat{r} - r^*\|_{L^2(\mathcal{S}, \ell^2)}^2$.

The first order optimality of $l(\cdot)$ implies $\nabla l(r^*) = 0$, then

$$l(r^*) - l(\hat{r}) = \nabla l(r^*)(r^* - \hat{r}) + \frac{1}{2}(r^* - \hat{r})^\top \nabla^2 l(\zeta)(r^* - \hat{r})$$
$$= 0 + \int_{\mathcal{S}} \frac{1}{N} \sum_{i<j} n_{ij} \frac{\partial^2}{\partial u^2} \log g(y; \zeta)(\hat{r}(s, a_i) - \hat{r}(s, a_j) - (r^*(s, a_i) - r^*(s, a_j)))^2 d\rho_s$$
$$\ge \int_{\mathcal{S}} \sum_{i<j} \frac{n_{ij}}{N} \kappa_2 (\hat{r}(s, a_i) - \hat{r}(s, a_j) - (r^*(s, a_i) - r^*(s, a_j)))^2 d\rho_s$$
$$\ge \kappa_2 \int_{\mathcal{S}} \kappa_\Lambda \sum_{a \in \mathcal{A}} (\hat{r}(s, a) - r^*(s, a))^2 d\rho_s$$
$$= \kappa_2 \kappa_\Lambda \|\hat{r} - r^*\|_{L^2(\mathcal{S}, \ell^2)}^2,$$

where there exists $y \in \Omega$ and $\zeta \in [\hat{r}(s, a_i) - \hat{r}(s, a_j), r^*(s, a_i) - r^*(s, a_j)]$. The second last step holds by the identifiability constraint for both the true and estimated reward function, i.e., $\sum_{a \in \mathcal{A}} \hat{r}(s, a) = 0$ and $\sum_{a \in \mathcal{A}} r^*(s, a) = 0$, which leads to $\sum_{a \in \mathcal{A}} (\hat{r}(s, a) - r^*(s, a)) = 0$. This identifiability constraint is also posited in (Shah et al., 2016). Therefore, we conclude that the final non-asymptotic error bound: with probability at least $1 - \delta$,

$$\|\hat{r} - r^*\|_{L^2(\mathcal{S}, \ell^2)}^2 \le 2\sqrt{2} \frac{\kappa_0}{\kappa_2 \kappa_\Lambda} \left( c_4 \sqrt{|\mathcal{A}|}(\lfloor\beta\rfloor + 1)^4 d^{\lfloor\beta\rfloor + 1}(\log(N))^2 N^{-\frac{\beta}{d+2\beta}} + \sqrt{\frac{\log(1/\delta)}{N}} \right).$$

Different from the deep regression and classification problem, the convergence of the excess risk does not directly ensure the function convergence of the estimated reward. It is necessary to consider additional constraints on the dataset structure (see Assumption 4). Assumption 4 ensures the Laplacian spectrum of the comparison graph to be bounded away from 0 and ensures the convergence of the estimator $\hat{r}$. The condition is also considered by Zhu et al. (2023). We note that to the best of our knowledge, we are the first to study its role in the non-parametric analysis for deep reward modeling.

□

# E  AN EXPERIMENT ON SYNTHETIC DATA

We construct a synthetic experiment to illustrate our theory in the guidance for deep neural network implementation. In this section, we consider two parametrization of the deep reward modeling using Bradley-Terry and Thurstonian models, respectively. We refer to the Example 1 and 2 for more details. This reward function is specified as: $r^*(s, a_1) = 2\sin(4\phi(s)^\top w^*)$ and $r^*(s, a_0) = -2\sin(4\phi(s)^\top w^*)$ where $\phi(s) = (\sin(s_1), \ldots, \sin(s_d))$ is a non-parametric transformation for creating non-linearity. The identification condition, $r^*(s, a_1) + r^*(s, a_0) = 0$ for every given $s$, is ensured. Furthermore, as demonstrated in expression (1) and (2), both regret and preference depend solely on the difference between rewards. Accordingly, in our implementation, we configure the output of the neural network to directly represent the reward difference, $\hat{r}(s, a_1) - \hat{r}(s, a_0)$, rather than estimating individual rewards separately.

We generate $n$ state observations $s$, with each sampled independently from a uniform distribution over $[0, 1]^d$. In this example, we consider each dataset with the dimension of $d = 10$ and sample size $(n_{train}, n_{eval}, n_{test}) = (2^{10}, 2^9, 2^{10})$. The true weight $w^*$ is fixed as $(-0.040, 1.726, -0.814, 1.372, 0.506, -0.482, -0.785, 0.668, -0.443, 0.188)^\top$, which is randomly generated *a priori*. We evaluate the rectangular neural networks where all the hidden layers are of the same width. Specifically, we consider the networks with widths $\{2^i, i = 4, \ldots, 12\}$ and the depths ranging from 3 to 13. The number of parameters ranges from about 500 to roughly $2 \times 10^7$. We note that the candidate networks have varying expression powers and are sufficient to validate the guidance ability of our theory in the network design. Each configuration is evaluated across 50 independent replications. The averaged regret results are presented in Figure 1.

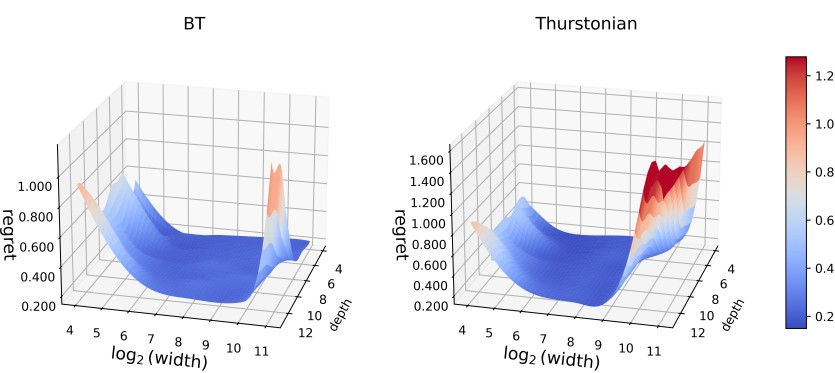

Figure 1: Regrets for synthetic data under different neural network configurations.

Recall that our theoretical analyses reveal a crucial trade-off in neural network architecture design. The interplay between stochastic and approximation errors fundamentally impacts model performance. Proposition 1 and Proposition 2 establish that while increased network complexity reduces approximation error, it simultaneously amplifies stochastic error under finite data scenarios. This finding emphasizes the importance of balanced architecture selection.

Our empirical results (see Figure 1) provide compelling evidence for this theoretical framework. Using a fixed sample size, we examined regret across varying network configurations. We observed a non-monotonic relationship: initially, deeper and wider networks reduce regret as the approximation capability of networks increases. However, beyond the near-optimal network configuration, further increases in model complexity led to degraded performance. This is consistent with our theory that stochastic error dominates under this case; this empirical evidence is in line with Han et al. (2023).

Furthermore, our results reveal that comparable performance levels can be achieved across diverse architectural configurations, highlighting the adaptability of deep neural networks to varying function complexities. This is evidenced by a relatively flat region in the parameter space where the regret remains near-minimal. Such architectural flexibility is particularly valuable in practice, as it suggests that precise knowledge of the smoothness parameter $\beta$ is not critical for achieving strong empirical performance. This robustness effectively addresses the common practical challenge of architecture selection under unknown function complexity. We recommend Jiao et al. (2023); Lee et al. (2019) for a more comprehensive analysis across different models.

