# OpenReview forum: "Learning guarantee of reward modeling using deep neural networks"
_ICLR.cc/2025/Conference — Submitted to ICLR 2025_

### Official Review · Reviewer_Yeyi · 2024-11-02

**Soundness:** 3
**Presentation:** 2
**Contribution:** 2
**Rating:** 5
**Confidence:** 3

**Summary:**

This work developed a regret bound (Thm.4) for RLHF when one approximates the true reward function with a deep neural network model (specifically, multi-layer perceptron) using MLE (Eq.(3)). It shows that having a margin condition (Assumption 1) can result in a faster rate (Thm.1 versus Cor 1) and these results could help readers identify better model architecture for reward modelling.

**Strengths:**

The paper provides a non-asymptotic regret bound through theoretical analysis, under assumptions on the preference margin (A1), finite true reward (A2) and some regularity conditions (A3).

**Weaknesses:**

1. The practicality of the result remains unclear. This work is mainly theoretical and there is no empirical evidence to show how useful the result would be. For example, L228 mentioned that “Our analysis provides implications for practitioners on how to choose the neural network parameters and construct high-quality comparison datasets to achieve effective reward modeling.” It would be much more convincing if the paper could provide some actual examples here. It is challenging to understand whether the required width and depth in Thm.4 are actually useful to design reward models and achieve better performance in practice.

2. The writing and presentation can be improved.

- It would be better to explain A1 using Examples 1 and 2. For example, what alpha and $c_g$ do we have in these two examples?
- L196 mentioned that this bound is “information-theoretically optimal”, but it remains unclear why.
- The overall structure of the paper is not easy to follow. Sec.2 looks more like an overlong overview of the whole paper, followed by Sec.3, which actually provides more technical details for the result in Sec.2. This is more common for long journal papers but does not work very well for a conference paper.

Minor

- L107: $r$ depends on $\pi_r$? Isn’t this reversed?
- L292: output of $H^{(i)}$ should be of the linear output?

**Questions:**

Q1. How useful is the main theorem in terms of practical algorithm design and achieving good empirical performance?

Q2. Why Thm.1 is “information-theoretically optimal”?

---

> ### Author Response · Authors · 2024-11-21
> **Response to Reviewer Yeyi (1/2)**
>
> Dear Reviewer,
>
> Thank you for your constructive comments and for dedicating your valuable time and effort toward the thorough evaluation of our paper. We have carefully considered your comments and the helpful suggestions provided. We have made significant efforts to address your concerns. Please see our point-by-point responses below.
>
> **Weakness 1: "about width and depth in design reward models"**
>
> Regarding the practicality of the result, we have conducted numerical experiments in our revision with an example for studying how the configuration of network architecture (depth and width) affects the regret of the learned estimators  (Please see Page 21 in the Appendix and the attached PDF in Supplementary Material in the rebuttal revision). Our simulation results show that the regret of the learned network estimations can achieve similarly low levels for a large region of values in the choices of the network depth and width as long as they are proper (not too small and not too large). In other words, the regret of the learned network estimations is robust to the choices of network width and depth. This observation is in line with our theory where proper order of network depth and width can lead to optimal rates. It offers a practical guideline for implementing our approach in real applications. Our empirical example also supports the theoretical guide regarding the importance of network depth. As illustrated, increasing the depth demonstrates superior performance gain compared to increasing the width with a similar number of parameters. Our findings on the efficiency of network depth and the robustness of network architecture choices (networks with different architectures achieving comparably good performance) have also been observed in the literature [1].
>
> **Weakness 2: "about writing and presentation"**
>
> Thank you for your suggestion.  In this revision, we have tried our best to improve the presentation of our manuscript based on your suggestions.
>
> (1) We would like to clarify that the noise exponent $\alpha$ describes the data quality, and $c_g$ is a constant that appears in the inequality, of which the exact value is unavailable. Both of them are determined by the data distribution but not tied to a specific parameterization of the preference model. Under this assumption, we derive $c^\prime_g$ for both Example 1 and Example 2 to make them adaptive to the inequality in Lemma 1. We would like to mention that Lemma 1 only depends additionally on the log-concavity of the comparison function $g$ and is thus applicable for various pairwise comparison models. We will include more details to improve its clarity in our revision.
>
> (2) We sincerely appreciate the reviewer’s questions. In this revision, we have updated the discussion near lines 187-190 to improve its accuracy. In this paper, we mean that the bounds are tight up to unknown constant factors by using the term "information-theoretically optimal." The "information-theoretically optimal" implies that the rate is optimal and cannot be improved further without additional conditions. We acknowledge that the terminology "information-theoretical optimality" is closely related to information theory [2,3].
>
> (3) Thank you for your helpful suggestion. In this revision, we have made our best efforts to improve the writing, especially in Sections 1 and 2, to make the paper structure more suitable for the conference.

---

> > ### Author Response · Authors · 2024-11-21
> > **Response to Reviewer Yeyi (2/2)**
> >
> > **Minor**:
> >
> > We appreciate your attention to these details and have made the necessary amendments: (1) The policy $\pi_{r}$ is determined by the reward estimate $r$. (2) $\sigma^{(i)}$ applies to the linear transformation $H^{(i)} x+b^{(i)}$ at the $i$-th layer. We have revised this issue.
> >
> > **Question 1: "about 'How useful is the main theorem in terms of practical algorithm design'"**
> >
> > Thank you for this important question about practical implications. As mentioned in ``weakness 1", we have conducted numerical experiments in our revision to provide insight into practical implications. In particular, we found that the regret of the learned network can be robust to the network architecture as long as the network depth and width have proper orders. In addition, our theoretical analysis provides theoretical justification for widely used data filtering and quality control mechanisms in RLHF implementations.  Particularly through the lens of the margin type condition, our theoretical framework explains why methods that focus on high-quality preference data and careful treatment of ambiguous samples achieve better performance - these approaches effectively improve the margin conditions identified in our analysis. It is also of our particular interest to study learning algorithms that are adaptive to this condition that could potentially accelerate preference learning in practice.
> >
> >
> > **Question 2: "about 'Why Thm.1 is information-theoretically optimal'"**
> >
> > Thank you for your insightful question. We refer to our answer to weakness 2 (2) for addressing this concern.
> >
> >
> > **Reference**
> >
> > [1] Han, J., Hu, M., & Shen, G. (2023). Deep neural newsvendor. arXiv preprint arXiv:2309.13830.
> >
> > [2] Jean-Yves Audibert and Alexandre B Tsybakov. Fast learning rates for plug-in classifiers. The Annals of Statistics, 35(2):608–633, 2007.
> >
> > [3] Bongole, R., Gouverneur, A., Rodríguez-Gálvez, B., Oechtering, T. J., & Skoglund, M. (2024). Information-Theoretic Minimax Regret Bounds for Reinforcement Learning based on Duality. arXiv preprint arXiv:2410.16013.

---

> > > ### Comment · Reviewer_Yeyi · 2024-11-23
> > >
> > > I thank the authors for the feedback.
> > > 1. It is great to have an empirical verification. However, the implementation details are not clear enough. For instance, it is assumed that the reward is normalized over actions (both $r^*$ and $\widehat{r}$ as discussed by Reviewer d2ZR). In Appendix E, $r^*$ is clearly not normalized while it is unknown whether $\widehat{r}$ is normalized.
> > > 2. Regarding information-theoretical optimality, I understand what it means and my question is *why* it "cannot be improved without additional conditions".

---

> > > > ### Author Response · Authors · 2024-11-24
> > > > **Response to Reviewer Yeyi**
> > > >
> > > > Dear Reviewer,
> > > >
> > > > We are grateful for the response! We have carefully considered your comments and the experiment has been updated in the Supplementary Materials. Please see our point-by-point responses below.
> > > >
> > > > **Comment 1:**
> > > >
> > > >
> > > > Thanks for your careful reading of our response and updated manuscripts.
> > > > First, we note that  it is necessary to impose the identification condition. Regarding normalization, the sum of $\hat r$, $r^*$  across the actions for given states can theoretically be set to 0 or any other constant without affecting the analysis.
> > > > Our theory is general and applies to any underlying reward function, not just normalized ones. For an unknown reward function, we can always transform it into a normalized version, and both the unnormalized and normalized functions lead to the same preference distribution. This ensures that our framework is robust to different reward function. For more details, we refer to Definition 1 and Lemma 1, 2 in [1]. They mention that reward functions $r(a,s), r^{\prime}(a,s)$ are equivalent if they satisfy $r(a,s)- r^{\prime}(a,s) = f(s)$ for some function $f(s)$. Such functions form an equivalence class and induce the same preference distribution.
> > > >
> > > > Lastly, we have updated the synthetic example in the Appendix to clarify this concept further. In this revision we consider the normalized underlying true reward function. Furthermore, as demonstrated in Expression (1) and (2), both the preference and regret depend solely on the difference between rewards. Accordingly, in our implementation, we configure the output of the neural network to directly represent the reward difference, $\hat{r}(s,a_1) - \hat{r}(s,a_0)$, rather than estimating individual rewards separately.
> > > >
> > > >
> > > >
> > > > **Comment 2:**
> > > >
> > > >
> > > > Thank you for reminding us about the discussion on Theorem 1. In the proof of Theorem 1, we only used Markov's inequality and selected $\eta$ to achieve the optimal convergence rate. Details can be found in Section C of the Appendix. The inequality bounds are tight, ensuring that our results are optimal and cannot be improved further without additional conditions.
> > > >
> > > > Notably, when  $\alpha=1$, the regret achieves the same rate as $||\hat r - r^* ||^2$, which aligns with the findings in the  [2]. In [2], the optimal fast rate is attained in the classification context under the margin-type condition and within the Holder function class. On page 148 of [2], it is further noted that classification can be treated as a special case of nonparametric estimation, which can be generalized. For a more detailed discussion, please refer to Theorem 1 and Section 4 of [2].
> > > >
> > > >
> > > > **Reference**
> > > >
> > > > [1] Rafailov, R., Sharma, A., Mitchell, E., Manning, C. D., Ermon, S., & Finn, C. (2024). Direct preference optimization: Your language model is secretly a reward model. Advances in Neural Information Processing Systems, 36.
> > > >
> > > > [2] Tsybakov, A. B. (2004). Optimal aggregation of classifiers in statistical learning. The Annals of Statistics, 32(1), 135-166.

---

> > > > > ### Author Response · Authors · 2024-11-27
> > > > > **Response to Reviewer Yeyi**
> > > > >
> > > > > Dear Reviewer Yeyi,
> > > > >
> > > > > Since the discussion phase ends soon, we wanted to ensure that our response has adequately addressed your concerns. We would be happy to address any remaining concerns or questions.
> > > > >
> > > > > Thank you for reviewing our work,
> > > > >
> > > > > Authors

---

> > > > ### Author Response · Authors · 2024-12-02
> > > >
> > > > **Dear Reviewer Yeyi**,
> > > >
> > > > As the discussion period nears its end, we kindly remind you that we are available to address any further questions you may have. We believe we have addressed all concerns. **If the reviewer is satisfied, we respectfully request that you consider raising your score before the deadline.** We sincerely thank you again for your time and efforts.

---

### Official Review · Reviewer_9CkQ · 2024-11-03

**Soundness:** 2
**Presentation:** 2
**Contribution:** 2
**Rating:** 5
**Confidence:** 2

**Summary:**

This paper proposes a new assumption on data, the margin-type condition, as a sufficient condition to achieve a sharp regret bound in reward modeling. The proposed assumption implies that experts exhibit a clear preference for the optimal action over others in most states, ensuring that the winning probability of the optimal action remains bounded away from 1/2. Under this assumption, the authors derive a tighter regret bound than existing results. Additionally, they establish regret bounds for reward modeling using deep neural networks, which explicitly depend on network structure and sample size.

**Strengths:**

- Formulating the intuitive idea that clear human feedback aids reward modeling into a margin-type condition was intriguing. As a result, the authors achieved a tighter regret bound for decision-making agents on data satisfying the margin-type condition. This outcome aligns with results in the contextual bandit literature, where margin conditions lead to tighter regret bounds [1, 2] .
-  Although the reviewer did not rigorously verify all proofs, the non-asymptotic excess risk bound for deep neural networks is also interesting. The authors provide theoretical guidance on the required network structure (width, depth) to attain a fast regret bound.

---
__Rerferences__

[1] Goldenshluger, Alexander, and Assaf Zeevi. "A linear response bandit problem." Stochastic Systems 3.1 (2013): 230-261.

[2] Bastani, Hamsa, and Mohsen Bayati. "Online decision making with high-dimensional covariates." Operations Research 68.1 (2020): 276-294.

**Weaknesses:**

- The paper seems to lack a sufficient literature review on non-asymptotic generalization error bounds for deep neural networks. There is insufficient discussion comparing the presented bound with prior works (e.g., [3]), identifying the challenges encountered, and clarifying the technical novelties.
- The neural network bound presented requires computation of an exact minimizer for the empirical loss (Eq. 3), which may be difficult to obtain in practice.

- - -
[3] Wang, Mingze, and Chao Ma. "Generalization error bounds for deep neural networks trained by sgd." arXiv preprint arXiv:2206.03299 (2022).

**Questions:**

1. What issues arise in the margin condition when $\alpha = 0$?

2. The structure of the neural network used to guarantee the results in Theorem 3 depends on $\beta$ (the Hölder class parameter). How can the $\beta$ of the reward model being estimated be determined? Is it assumed to be prior knowledge?

**Details Of Ethics Concerns:**

There are no specific ethics concerns.

---

> ### Author Response · Authors · 2024-11-21
> **Response to Reviewer 9CkQ**
>
> Dear Reviewer,
>
> We appreciate your thorough review of our paper and your positive feedback on our work. We have carefully considered your comments and have made significant efforts to address each of your concerns. Please see our point-by-point responses below.
>
> **Weakness 1: "about insufficient review on error bounds for dnn"**
>
> Thank you for your suggestion and for bringing these important studies ([1,2,3]) to our attention. In this revision, we have acknowledged the valuable theoretical studies referenced above. Specifically, we have included a literature review in Section 4 that focuses on the convergence analysis of deep neural networks, thereby enhancing the comprehensiveness of our discussion.
> In terms of technical novelties, we emphasized them in the revised version and summarized them as follows:
> First, unlike typical deep regression learning, our context involves an additional identification problem in reward modeling that needs to be carefully addressed. In the revised version, we emphasize the identification constraint in Lines 276-279 and the corresponding theoretical result in Theorem 3 and 4.
> Second, beyond the deep neural network structure, our proof considers the influence of the underlying structure of the pairwise comparison dataset. Specifically, we explicitly illustrate how the comparison data impacts the convergence rate via the data coverage assumption. This point is emphasized in Assumption 4 and Theorem 3. To the best of our knowledge, our analysis is the first to explore deep reward modeling under data coverage conditions.
> Last but not least, our theory reveals the connection between the intuitive notion of clear human beliefs and the reward gap between the optimal action and others. Specifically, we link clear preferences to the probability gap in the dataset rather than directly imposing assumptions on the underlying reward function. This provides a comprehensive explanation of how the margin-type condition contributes to the final convergence rate in reward modeling. Additionally, it offers a theoretical approach to formalizing empirical insights about clear preferences in RLHF modeling.
>
> **Weakness 2: "about the optimization error"**
>
> We thank the reviewer for providing the insightful work and helping to improve the quality of this paper.  We agree that optimization error is a very important but challenging one, which is of independent interest to study [1].  We acknowledge that the stochastic and approximation errors are studied, while the optimization error is not addressed in this paper and we will ensure that this is clearly stated in our revision to provide clarity for the readers. In the revision, we include the relevant discussion based on your provided literature to highlight the importance of optimization error in the theoretical analysis of the DNN estimator.
>
> **Question 1: "about the margin condition when α=0"**
>
> $\alpha=0$ implies no additional assumption is made on the data distribution, allowing the existence of greater ambiguity in the preference dataset. In this case, the regret bound defaults to the slow rate, and any policy $\pi_{r}$ may require more data to achieve the same level of accuracy compared to situations with α>0.
>
> **Question 2: "about unknown smoothness parameter."**
>
> Thank you very much for your helpful questions. In practice, the smoothness parameter may not be directly inferred. However, many empirical experiments validate that deep networks yield good performance and can cover the most commonly considered function class.
> In this revision, we provide a toy example to study the relationship between the performance and network parameters. The details of the experiments are attached in the Appendix. As shown, a relatively flat region corresponds to (nearly) minimum excess risk. It implies that it is unnecessary to calculate the network structure with the smoothness parameter explicitly.  Many network structures can achieve satisfactory numerical performance.
>
> **Reference**
>
> [1] Wang, Mingze, and Chao Ma. "Generalization error bounds for deep neural networks trained by sgd." arXiv preprint arXiv:2206.03299 (2022).
>
> [2] Goldenshluger, Alexander, and Assaf Zeevi. "A linear response bandit problem." Stochastic Systems 3.1 (2013): 230-261.
>
> [3] Bastani, Hamsa, and Mohsen Bayati. "Online decision making with high-dimensional covariates." Operations Research 68.1 (2020): 276-294.

---

> > ### Author Response · Authors · 2024-11-27
> > **Response to Reviewer 9CkQ**
> >
> > Dear Reviewer 9CkQ,
> >
> > Since the discussion phase ends soon, we wanted to ensure that our response has adequately addressed your concerns. We would be happy to address any remaining concerns or questions.
> >
> > Thank you for reviewing our work,
> >
> > Authors

---

> > ### Author Response · Authors · 2024-12-02
> > **Response to Reviewer 9CkQ**
> >
> > **Dear Reviewer 9CkQ**,
> >
> > As the discussion period nears its end, we kindly remind the reviewer that we are available to address any further questions they may have. We believe we have addressed the concerns. Specifically,
> >
> > -  Following your advice, we have added the literature review for the generalization analysis of deep learning.
> >
> > - We have incorporated the important literature you mentioned into the discussion to make the paper more comprehensive.
> >
> > - We have revised the manuscript to clarify the details you highlighted.
> >
> >  **If the reviewer is satisfied, we respectfully request that they consider raising their score before the deadline.** We sincerely thank them again for their time and efforts.

---

### Official Review · Reviewer_bWGB · 2024-11-07

**Soundness:** 3
**Presentation:** 3
**Contribution:** 3
**Rating:** 6
**Confidence:** 2

**Summary:**

This paper introduces a theoretical framework for understanding reward modeling in reinforcement learning (RL) based on pairwise comparisons and fully connected deep neural networks (DNNs) to estimate the reward function. When fine-tuning large language models, previous studies have used pairwise human feedback data to model rewards, enhancing sample efficiency and performance alignment with human preferences. The authors develop a non-asymptotic regret bound for DNN-based reward estimators, providing learning guarantees that explicitly account for neural network architecture parameters (i.e., width and depth). In Section 2, the authors develop learning guarantees under margin-type conditions, which ensure a significant margin in preference probabilities (i.e., the winning probability for an optimal action in comparisons) is well-separated from randomness. This condition enables sharper regret bounds. This result emphasizes the role of high-quality pairwise data in achieving efficient Reinforcement Learning from Human Feedback (RLHF) outcomes. In Section 3, the authors derive regret bounds that depend on DNN structure, demonstrating the role of network depth, width, and sample size in achieving high sample efficiency.

**Strengths:**

1. Theoretical results are “fine-grained,” as they consider specific neural network structures rather than relying on generalized assumptions about network properties.
2. Moreover, the paper addresses both stochastic and approximation error bounds, and provides guidance on achieving an optimal balance between these by designing the width and depth of DNNs. This is a nice attempt to bridge the theory to real-world model design.

**Weaknesses:**

1. The claimed extension (line 378-381) from DNNs to state-of-the-art architectures (such as BERT or GPT) is not fully convincing. While functionally similar, these architectures differ significantly in pipelines, loss design, training methods, and especially in their use of attention mechanisms and transformer layers, which are not addressed in your analysis. Consequently, I believe the gap between these theoretical results and the practical guidance needed for fine-tuning in RLHF remains significant. It would strengthen your claim if you could specifically address how these differences impact the applicability of your results or provide insights on adapting your analysis to accommodate these architectural and training nuances.

2. Even if we adopt the margin-type condition to data collection in practice, I am not sure how applicable it would be. Sometimes we encounter ambiguous or complex target data where a “clear human belief” is not always possible. Neither enforcing a clear preference nor ignoring the data point after identifying its difficulty is a good solution.

**Questions:**

1. Are there any toy experiments or future studies the authors can think of that would help validate the theoretical framework and findings? For example, could any experiments demonstrate the optimality of the claimed model depth and width, in terms of balancing stochastic and approximation errors?

2. Related to the Weakness point 1, are there any versions of Theorem 4 that might extend beyond ReLU fully connected networks? Perhaps adaptations for attention mechanisms? Note that this possiblity is explicitly mentioned in line 378 - 381.

---

> ### Author Response · Authors · 2024-11-21
> **Response to Reviewer  bWGB (1/2)**
>
> Dear Reviewer,
>
> We appreciate your thorough review of our paper and your positive feedback on our work. We have carefully considered your comments for improvement. Please see our point-by-point responses below.
>
> **Weakness 1: "about possibilities to extend to other network structures"**
>
> Thank you for sharing your concern regarding the network structure. We acknowledge the gap between these theoretical results and the practical guidance for fine-tuning in reinforcement learning with human feedback (RLHF). The claimed extension mainly considers the reduced complexity of neural networks through the lens of functional equivalence [1]. Such functional equivalence generally exists for fully connected networks, Residual networks, and attention-based networks and it is possible to extend the theoretical convergence analysis for the fully connected networks considered in this paper to other network structures via the functional equivalence approach.  [2] derived the convergence rate for transformer networks using a similar analysis based on the stochastic error and the approximation error. And Theorem 5.4 of [2] achieves a polynomial convergence rate, akin to that found in ReLU network analysis.  Notably, they also consider the shift-equivariant properties of transformer networks, which supports this claim.
> In this revision, we have acknowledged and discussed the connection and gap between the fully connected neural networks and attention-based networks and recommend [3,4,5] for more comprehensive theoretical studies of transformers in Lines 391-399.
>
> **Weakness 2: "about the applicability of the margin type condition"**
>
> We appreciate your insightful comment about the practical applicability of the margin-type condition. Our results provide a precise characterization of RHLF's performance across different scenarios, explaining both sample efficiency and potential limitations up to a margin exponent. Even in cases where the margin condition is not fully satisfied (that is $\alpha = 0$), our analysis shows convergence is still achievable, albeit requiring larger sample sizes and more careful model design. Previous literature ([6,7,8]) has demonstrated valuable practical implementations. For example, [6] mentioned that in the fine-tuning of GPT-3, training labelers agree with each other 72.6% of the time on average despite the complexity of the task, which is quite high. It implies that the “clear human belief”  may be more common in practice. [7] proposed using contrastive learning methods to differentiate the data embedding according to preference labels. [8] invokes active learning algorithms to conduct context selection in RLHF.
> We refer to Section 4 in the main text of our paper for more discussion on recent empirical studies that have developed various strategies for managing crowd-sourced preference data with varying levels of ambiguity. Our theoretical results support these practical implementations, such as enhancing data quality and handling ambiguous samples. As the reviewer suggests, while this approach may not always be the optimal solution, it is a valuable strategy in practice. We recommend the literature in discussions on related work for more details.

---

> ### Author Response · Authors · 2024-11-21
> **Response to Reviewer bWGB (2/2)**
>
> **Question 1: "about the network design analysis guiding empirical implementation"**
>
> Thank you very much for your helpful questions. In this revision, we have included an example of a simulation study to demonstrate the relationship between network architecture and regret. The experimental details are provided in the Appendix.
> In particular, our simulation results show that the regret of the learned network estimations can achieve similarly low levels for a large region of values in the choices of the network depth and width as long as they are proper (not too small and not too large). In other words, the regret of the learned network estimations is robust to the choices of network width and depth. This observation is in line with our Theorem 4 where proper order of network depth and width can lead to optimal rates. It offers a practical guideline for implementing our approach in real applications. In addition, our empirical example supports the theoretical guide regarding the importance of network depth. As illustrated in our fixed sample size setting, increasing the depth demonstrates superior performance gain compared to increasing the width, although both yield a similar incremental number of network parameters. Our findings on the efficiency of network depth and the robustness of network architecture choices (networks with different architectures achieving comparably good performance) have also been observed in the literature [9].
>
> **Question 2: "about the extension beyond ReLU networks"**
>
> Regarding the extension of Theorem 4 beyond ReLU fully connected networks, we acknowledge the potential for such adaptations, including the attention mechanisms. Recent studies [2,3,4,5] collectively underscore the robust theoretical underpinnings and wide applicability of Transformers and attention mechanisms, supporting the potential extensions and adaptations mentioned. We have included more discussions to mention the potential for future studies in the main text.
>
> **Reference**
>
> [1] Shen, G. (2024). Complexity of deep neural networks from the perspective of functional equivalence. In International Conference on Machine Learning, 41, 2024.
>
> [2] Takakura, S., & Suzuki, T. (2023, July). Approximation and estimation ability of transformers for sequence-to-sequence functions with infinite dimensional input. In International Conference on Machine Learning (pp. 33416-33447). PMLR.
>
> [3]  Deora, P., Ghaderi, R., Taheri, H., & Thrampoulidis, C. (2023). On the optimization and generalization of multi-head attention. arXiv preprint arXiv:2310.12680.
>
> [4] Yun, C., Bhojanapalli, S., Rawat, A. S., Reddi, S. J., & Kumar, S. (2019). Are transformers universal approximators of sequence-to-sequence functions?. arXiv preprint arXiv:1912.10077.
>
> [5] Fang, Z., Ouyang, Y., Zhou, D. X., & Cheng, G. (2022). Attention enables zero approximation error. arXiv preprint arXiv:2202.12166.
>
> [6] Ouyang, L., Wu, J., Jiang, X., Almeida, D., Wainwright, C., Mishkin, P., ... & Lowe, R. (2022). Training language models to follow instructions with human feedback. Advances in neural information processing systems, 35, 27730-27744.
>
> [7] Wang, B., Zheng, R., Chen, L., Liu, Y., Dou, S., Huang, C., ... & Jiang, Y. G. (2024). Secrets of rlhf in large language models part ii: Reward modeling. arXiv preprint arXiv:2401.06080.
>
> [8] Das, N., Chakraborty, S., Pacchiano, A., & Chowdhury, S. R. (2024). Active preference optimization for sample efficient RLHF. In ICML 2024 Workshop on Theoretical Foundations of Foundation Models.
>
> [9] Han, J., Hu, M., & Shen, G. (2023). Deep neural newsvendor. arXiv preprint arXiv:2309.13830.

---

> > ### Comment · Reviewer_bWGB · 2024-11-21
> >
> > Thanks for the authors' response. Overall, I think this paper produces novel theoretical results and insights, which are relevant to the LLM fine-tuning practices. After consideration, I have raised my rating.

---

> > > ### Author Response · Authors · 2024-11-22
> > > **Response to the Comment of Reviewer bWGB**
> > >
> > > We sincerely thank the reviewer for their thoughtful review and appreciation of our work, which has contributed greatly to improving this paper.

---

### Official Review · Reviewer_HieT · 2024-11-08

**Soundness:** 2
**Presentation:** 2
**Contribution:** 2
**Rating:** 6
**Confidence:** 3

**Summary:**

The paper theoretically studies reward modeling using pairwise comparison
data and deep neural networks. They obtained a regret bound for deep neural network reward estimators which explicitly depends on network architectures such as width and depth. Moreover, they introduce a margin-type condition that measures the confidence of the human belief. This margin-type condition enables a sharper regret bound, which improves the regret bound from $\| r-r^*\|^{2/3}$ to  $\| r-r^*\|^2$ in the most extreme case.

**Strengths:**

1. The paper studies the theory of reward modeling, which is a very important question for LLMs.
2. The paper provides regret bound for neural network structures which is used in practice.
3. The paper introduced the margin condition which can quantify the confidence of the human preference which does not rely on the underlying reward model. The paper then obtained a sharper reward bound given the margin condition.

**Weaknesses:**

1. Although the theory looks solid, there is no/few surprise or new insights provided in this paper.
2. Arguably, the most important contribution in this paper is introducing the margin condition which can quantify the confidence of the human preference. However, an empirical verification of this assumption in real-world datasets is missing.

**Questions:**

Does example 1 or 2 satisfy assumption 1? If so, what is the corresponding alpha?

---

> ### Author Response · Authors · 2024-11-21
> **Response to Reviewer HieT (1/2)**
>
> Dear Reviewer,
>
> Thank you for your overall positive comments and for dedicating your valuable time and effort toward the thorough evaluation of our paper. We have carefully considered your comments and made significant efforts to address each of your concerns. Please see our point-by-point responses below.
>
> **Weakness 1: "about new insights provided in this paper"**
>
> Thank you for your comments. We would like to highlight several key aspects that distinguish ours from conventional ones. We have emphasized these differences in the revised version and summarized them as follows:
> First, we model clear human beliefs through the selection probability gap between optimal actions and others. Unlike many existing studies, we link clear preferences to the probability gap in the dataset rather than directly imposing assumptions on the underlying reward function. To the best of our knowledge, we are the first to model clear human beliefs using a margin-type condition in Reinforcement Learning from Human Feedback (RLHF).
> Second, our result incorporates the influence of the underlying structure of the pairwise comparison dataset. Based on the proposed data coverage assumption, we explicitly demonstrate how the comparison data affects the convergence rate. This point is emphasized in Assumption 4 and Theorem 3. To the best of our knowledge, our analysis is the first to address deep reward modeling in the context of data coverage conditions.
> Third, unlike typical deep regression learning, we carefully addressed the additional identification problem in reward modeling in the context of the comparison model and RLHF. In the revised version, we emphasize the identification constraint in Line 276-278 and the corresponding theoretical result in Theorem 3 and 4.
>
> **Weakness 2: "about verification of this margin assumption"**
>
> Thank you for sharing your concern regarding the margin-type condition. In this paper, we focus on developing a theoretical understanding of clear human beliefs based on the margin-type condition as many existing studies (listed as follows) emphasize the importance of clear preferences and suggest various approaches for handling ambiguous samples in practice. Our work introduces the margin-type condition as a way to model clear human preferences in Reinforcement Learning from Human Feedback (RLHF) to mitigate the gap in its theoretical understanding. To the best of our knowledge, this is the first theoretical quantification of high-quality data in RLHF.
> Here we list some existing studies that implicitly mention the importance of this assumption in real-world datasets. [1] mentioned that in the fine-tuning of GPT-3, training labelers agree with each other 72.6% of the time on average despite the complexity of the task, which is quite high. [2] mentioned that the presence of incorrect and ambiguous preferences in the dataset due to the low agreement among annotators during preference labeling could lead to difficulties in accurately representing human preferences. They suggest that detecting and mitigating noisy data is essential for aligning learned rewards with true human preferences.
> Recently, [3] has given empirical results comparing the performance by training the model with (i) only the clear preference samples and (ii) both clear preference samples and tied pairs. They conclude that including tied pairs in direct preference optimization is not good for task performance. A possible solution includes modeling the ties explicitly ([3]), correcting the labels of wrong preferences, and smoothing the labels of ambiguous preferences to avoid the model’s overfitting on these low-quality data, as suggested by [4].

---

> ### Author Response · Authors · 2024-11-21
> **Response to Reviewer HieT (2/2)**
>
> **Question: "Does example 1 or 2 satisfy assumption 1"**
>
> Thanks for your questions. We would like to clarify that the noise exponent $\alpha$ is not associated with any specific pairwise comparison probability model.  Instead, this assumption pertains to the state (data) distribution and is intended to describe the quality of data obtained from pairwise comparisons. We have acknowledged this point in the main text and provided details in Example 4 and 5 in the Appendix to make it clearer.
>
>
> **Reference**
>
> [1] Ouyang, L., Wu, J., Jiang, X., Almeida, D., Wainwright, C., Mishkin, P., ... & Lowe, R. (2022). Training language models to follow instructions with human feedback. Advances in neural information processing systems, 35, 27730-27744.
>
> [2] Wang, B., Zheng, R., Chen, L., Liu, Y., Dou, S., Huang, C., ... & Jiang, Y. G. (2024). Secrets of rlhf in large language models part ii: Reward modeling. arXiv preprint arXiv:2401.06080.
>
> [3] Chen, J., Yang, G., Lin, W., Mei, J., & Byrne, B. (2024). On Extending Direct Preference Optimization to Accommodate Ties. arXiv preprint arXiv:2409.17431.
>
> [4] Zheng, R., Xi, Z., Liu, Q., Lai, W., Gui, T., Zhang, Q., ... & Ge, W. (2023, July). Characterizing the impacts of instances on robustness. In Findings of the Association for Computational Linguistics: ACL 2023 (pp. 2314-2332).

---

> > ### Comment · Reviewer_HieT · 2024-11-26
> >
> > Thank you for your response. I appreciate the clarification from the authors and will keep my score.

---

> > > ### Author Response · Authors · 2024-11-27
> > > **Response to Reviewer HieT**
> > >
> > > We again thank the reviewer for their thoughtful review and appreciation of our work, which has contributed greatly to improving this paper.

---

### Official Review · Reviewer_d1ZR · 2024-11-09

**Soundness:** 4
**Presentation:** 3
**Contribution:** 3
**Rating:** 6
**Confidence:** 4

**Summary:**

The paper gives a regret analysis of reward model learning that incorporates
1. An upper bound of average reward regret in terms of the L_2 functional error of the learned reward function, under a margin condition.
2. An L_2 functional error guarantee of the maximal-likelihood neural network reward function solution with the holder-smooth realizability condition.

**Strengths:**

The paper is clearly written and easy to follow. That includes
1. clear illustrations of the relationship between regret, functional error of the reward model, and the maximal-likelihood solution
2. clear statements of the assumptions
3. clear proofs
The paper provides a good characteristic of the reward signal distribution that go beyond the BT models.

**Weaknesses:**

1. The assumptions in the paper may be restrictive, namely the realizability of the reward model (the existence of an underlying model) and the data coverage assumption (the 2nd smallest eigenvalue of the data coverage Laplacian). This happens to not to match current practice where the signal is sparse and there may not be a true reward model.

2. Many proofs in the paper seem standard in the literatures, especially the generalization analysis of holder-smooth neural networks. That reduces the contribution in novelty of the whole paper.

**Questions:**

- A theoretical concern: is the inequality in line 960 only true when $\sum_{a}\hat{r}(s,a)-r^*(s,a)=0$? If so I cannot see why this is true.

Suggestions:
1. There are several links in the informal theorems that causes confusion to readers (e.g. links in theorem 2 to assumptions in the later part of the paper). It would be favorable if the assumptions can be described. Also Definition 1 should be ahead of Lemma 1 as it is required there.
2. The 2nd smallest eigenvalue should be proposed as an assumption in the paper to make theorem 4 comprehensible.

---

> ### Author Response · Authors · 2024-11-21
> **Response to Reviewer d1ZR (1/2)**
>
> Dear Reviewer,
>
> Thank you for your overall positive assessment of our paper and for dedicating your valuable time and effort toward the thorough review. We have carefully considered your comments and have made significant efforts to address each of your concerns. Please see our point-by-point responses below.
>
> **Weakness 1: "about the assumption on the existence of an underlying reward model and the data coverage assumption"**
>
> Thanks for the comments about our assumptions on the reward model and data coverage. We would like to make the clarification as follows.
> First, the assumption on the existence of a reward model comes from the general reinforcement learning problems [1], especially reinforcement learning with human feedback (RLHF) [2,3,4,5] and inverse reinforcement learning [6]. Since our work studies the learning guarantees of reward modeling in the RLHF, it is unavoidable that we should adopt this assumption. It is still underexplored to model human preference without an underlying golden reward model. We anticipate if there is another systematic approach to explain human preference and be willing to study its theoretical performance in the future.
> Second, the data coverage assumption is a necessary assumption to guarantee the converging of the maximum likelihood estimator (MLE), which is the most widely used approach in RLHF[3,7]. Specifically, [7] and [8] show that the MLE will fail if the dataset is insufficiently covered. In this work, we further explore how the coverage of the data influences the convergence rate of the MLE. Our result in Theorem 3 reveals the error bound will not be controlled once the 2nd smallest eigenvalue is sufficiently small. How to deal with insufficient coverage issues and build the corresponding learning guarantee are very interesting but challenging problems that deserve future studies.
>
> **Weakness 2: "about the novelty of the proofs"**
>
> We understand your concern about the novelty of our proofs given the general framework of error decomposition we have used for establishing theoretical results in deep learning. Nonetheless, we would like to highlight three key aspects that differentiate our proof from the standard approach. These have been emphasized in the revised version and are summarized as follows:
> First, in addition to the deep neural network structure, our proof incorporates the influence of the underlying structure of the pairwise comparison dataset. Based on the proposed data coverage assumption, we explicitly demonstrate how the comparison data affects the convergence rate. This point is emphasized in Assumption 4 and Theorem 3. To the best of our knowledge, our analysis is the first to address deep reward modeling in the context of data coverage conditions.
> Second, our theory reveals the connection between the intuitive notion of clear human beliefs and the reward gap between the optimal action and others. Specifically, we link clear preferences to the probability gap in the dataset rather than directly imposing assumptions on the underlying reward function. This provides a comprehensive explanation of how the margin-type condition contributes to the final convergence rate in reward modeling. Additionally, it offers a theoretical approach to formalizing empirical insights about clear preferences in RLHF modeling.
> Lastly, unlike typical deep regression learning, our situation involves the additional identification problem in reward modeling, which requires careful addressing. In the revised version, we emphasize the identification constraint in Lines 276-279 and the corresponding theoretical result in Lines 1073-1076.

---

> > ### Author Response · Authors · 2024-11-21
> > **Response to Reviewer d1ZR (2/2)**
> >
> > **Question 1: "about inequality in line 960"**
> >
> > Thanks for your careful reading of our proof. Yes, this step uses the equality that $\sum_{a\in \mathcal{A}}( \hat{r}(s,a) - r^*(s,a))=0$. We would like to clarify that this is a commonly-used constraint on both the true and estimated reward function where $\sum_{a\in \mathcal{A}} r^*(s,a)=0$ and $\sum_{a\in \mathcal{A}}\hat{r}(s,a)=0$ for any state $s$. This constraint guarantees that the reward is identifiable and is commonly adopted in literature such as [7]. We highlight this constraint in the revised paper to make it clearer.
> >
> >
> > **Suggestion 1: "the rearrangement of links in Section 2"**
> >
> > Thanks for your suggestions. Based on your feedback, we have reorganized the links and included detailed descriptions for the definitions and assumptions in the revised version. We believe these changes have made the manuscript clearer and easier to follow.
> >
> > **Suggestion 2: "make data coverage condition an assumption"**
> >
> > Thanks for your suggestions. In the revised version, we have formulated the data coverage condition as an assumption and emphasized its importance in theoretical analysis. We refer to Assumption 4 in the main text and Lines 1080-1086 in the appendix for more details.
> >
> >
> > **Reference**
> >
> > [1] Sutton, R. S. (2018). Reinforcement learning: An introduction. A Bradford Book.
> >
> > [2] Christiano, P. F., Leike, J., Brown, T., Martic, M., Legg, S., & Amodei, D. (2017). Deep reinforcement learning from human preferences. Advances in neural information processing systems, 30.
> >
> > [3] Rafailov, R., Sharma, A., Mitchell, E., Manning, C. D., Ermon, S., & Finn, C. (2024). Direct preference optimization: Your language model is secretly a reward model. Advances in Neural Information Processing Systems, 36.
> >
> > [4] Wang, B., Zheng, R., Chen, L., Liu, Y., Dou, S., Huang, C., ... & Jiang, Y. G. (2024). Secrets of rlhf in large language models part ii: Reward modeling. arXiv preprint arXiv:2401.06080.
> >
> > [5] Song, F., Yu, B., Li, M., Yu, H., Huang, F., Li, Y., & Wang, H. (2024, March). Preference ranking optimization for human alignment. In Proceedings of the AAAI Conference on Artificial Intelligence (Vol. 38, No. 17, pp. 18990-18998).
> >
> > [6] Fu, J., Luo, K., & Levine, S. (2018, February). Learning Robust Rewards with Adverserial Inverse Reinforcement Learning. In International Conference on Learning Representations.
> >
> > [7] Zhu, B., Jordan, M., & Jiao, J. (2023, July). Principled reinforcement learning with human feedback from pairwise or k-wise comparisons. In International Conference on Machine Learning (pp. 43037-43067). PMLR.
> >
> > [8] Shah, N. B., Balakrishnan, S., Bradley, J., Parekh, A., Ramch, K., & Wainwright, M. J. (2016). Estimation from pairwise comparisons: Sharp minimax bounds with topology dependence. Journal of Machine Learning Research, 17(58), 1-47.

---

> > > ### Author Response · Authors · 2024-11-27
> > > **Response to Reviewer d1ZR**
> > >
> > > Dear Reviewer d1ZR,
> > >
> > > Since the discussion phase ends soon, we wanted to ensure that our response has adequately addressed your concerns. We would be happy to address any remaining concerns or questions.
> > >
> > > Thank you for reviewing our work,
> > >
> > > Authors

---

> > > > ### Comment · Reviewer_d1ZR · 2024-11-27
> > > >
> > > > Thank you for your reply. Yeah you addressed all the concerns, and I think the paper is good for a rating above the accepting threshold.

---

> > > > > ### Author Response · Authors · 2024-11-27
> > > > >
> > > > > Dear Reviewer d1ZR,
> > > > >
> > > > >  we again thank you for your thoughtful review and appreciation of our work, which has contributed greatly to improving this paper.

---

### Author Response · Authors · 2024-11-27
**Comments by the Authors of 8930**

Dear Reviewers,

 As the author-reviewer discussion period is ending soon, we kindly ask if you could review our responses to your comments. If you have further questions or comments, we will do our best to address them before the discussion period ends. If our responses have resolved your concerns, we would greatly appreciate it if you could update your evaluation of our work accordingly.

---

### Author Response · Authors · 2024-12-02

**Dear Reviewers**,

First and foremost, we express our sincere gratitude to the five reviewers for their invaluable feedback, which has helped us improve the quality of our manuscript. We have revised our manuscript, taking all the reviewers' comments into consideration. As the author-reviewer discussion period is nearing its end, we kindly request your review of our responses to your comments. Below, we summarize the major changes made to address the primary concerns raised by the reviewers.

- We highlight our technical contributions relative to existing works and further discuss the implications of our main results.

- We add a synthetic example to illustrate the implications of our results in designing deep neural network structures for reward modeling.

- We include a literature review on the generalization theory in deep learning following the comments of the reviewer.

- We add discussions for some numerical evidence for the margin-type condition and the potential for extending the result on many state-of-the-art Transformer networks.


We have highlighted the changes in the revision.

Authors of Submission 8930

---

### Meta-Review · Area_Chair_WrNx · 2024-12-23

**Metareview:**

This paper presents a set of theoretical guarantees for learning reward from pair-wise data and comparison feedback. While the set of results are rather comprehensive, covering a diverse set of types and settings. Its technical significance, contribution to new insights, and connection to practice remains not sufficient for ICLR. We thus recommend rejection.

**Additional Comments On Reviewer Discussion:**

Positive reviewers are not convinced with technical contribution, negative reviewers questioning the practical relevance of the results. Both contributes to the decision of rejection of this paper.

---

### Decision · Program_Chairs · 2025-01-22

Reject